# The response of stratospheric water vapor to climate change driven by different forcing agents

Xun Wang[1] and Andrew E. Dessler[1]

[1]Department of Atmospheric Sciences, Texas A&M University, College Station, TX, USA

*Correspondence to*: Andrew E. Dessler (adessler@tamu.edu)

**Abstract.** We investigate the response of stratospheric water vapor (SWV) to different forcing agents within the Precipitation Driver and Response Model Intercomparison Project (PDRMIP) framework. For each model and forcing agent, we break the SWV response into a slow response, which is coupled to surface temperature changes, and a fast response, which is the response to external forcing, but before the sea surface temperatures have responded. Our results show that, for

most climate perturbations, the slow SWV response dominates the fast response. The slow SWV response exhibits a similar sensitivity to surface temperature across all climate perturbations. Specifically, the sensitivity is 0.35 ppmv K$^{-1}$ in the tropical lower stratosphere (TLS), 2.1 ppmv K$^{-1}$ in the northern hemispheric lowermost stratosphere (LMS), and 0.97 ppmv K$^{-1}$ in the southern hemispheric LMS. In the TLS, the fast SWV response only dominates the slow SWV response when the forcing agent radiatively heats the cold point region — for example, black carbon, which directly heats the atmosphere by absorbing

solar radiation. The fast SWV response in the TLS is primarily controlled by the fast adjustment of cold point temperature across all climate perturbations. This control becomes weaker at higher altitudes in the tropics and altitudes below 150 hPa in the LMS.

## 1 Introduction

Stratospheric water vapor plays an important role in global climate change. It is an important greenhouse gas (GHG), which

affects the Earth's radiative budget (Forster and Shine, 2002; Solomon et al., 2010), and it also plays an important role in stratospheric ozone chemistry (Solomon et al., 1986; Dvortsov and Solomon, 2001).

SWV in the overworld (above 380-K isentropic surface) (e.g. Hoskins, 1991) and SWV in the extratropical lowermost stratosphere (LMS, between the extratropical tropopause and the 380-K isentropic surface) (e.g. Holton et al., 1995) are distinguished according to different mechanisms that control them. Overworld SWV is primarily controlled by the

temperatures in the tropical tropopause layer (TTL) as air is transported through it (e.g. Mote et al., 1996; Fueglistaler et al., 2009) and by production from oxidation of methane (e.g. Brasseur and Solomon, 2005). The LMS SWV is controlled by three major sources, including the transport of overworld air by the downward branch of Brewer-Dobson circulation, adiabatic quasi-horizontal transport from the tropical upper troposphere, and diabatic cross-tropopause transport due to deep

convection (Dessler et al., 1995; Holton et al., 1995; Plumb, 2002; Gettelman et al., 2011).

The response of SWV to climate change can be partitioned into two components: the fast response and slow response. The addition of a radiatively active constituent to the atmosphere can influence the atmosphere even before the surface temperature changes, leading to changes in SWV. This is often referred to as an "adjustment" to the forcing, and is generally considered part of the external forcing (e.g. Sherwood et al., 2015). We will refer to this as the "fast response" of SWV to the forcing. The slow response is the component in the SWV change that is coupled to changes of the surface temperature,

which occurs on longer time scales. This slow response means that SWV could be an important positive feedback to global warming (Forster and Shine, 2002; Dessler et al., 2013; Huang et al., 2016; Banerjee et al., 2019). Banerjee et al. (2019) have shown that, when $CO_2$ is abruptly quadrupled, the change in SWV mainly consists of the slow response and that the fast response is less important.

Previous studies have shown that climate models, which are able to accurately reproduce observed interannual variations in

SWV (Dessler et al., 2013; Smalley et al., 2017), robustly project a positive long-term trend in overworld SWV at entry level with a warming climate due to increasing GHGs (Gettelman et al., 2010; Dessler et al., 2013; Smalley et al., 2017). This is mainly due to a warmer tropopause (Thuburn and Craig, 2002; Gettelman et al., 2010; Lin et al., 2017; Smalley et al., 2017; Xia et al., 2019), which is controlled, to some extent at least, by the warming surface (Gettelman et al., 2010; Shu et al., 2011; Dessler et al., 2013; Huang et al., 2016; Revell et al., 2016; Lin et al., 2017; Smalley et al., 2017; Banerjee et al.,

2019). Dessler et al. (2016) suggested that increases in convective injection into the stratosphere due to a warming climate may also be contributing to the trend in entry SWV. In the LMS, the climate models show larger increases in SWV (Dessler et al., 2013; Huang et al., 2016; Banerjee et al., 2019). It is not known how SWV responds to different forcing agents. Hodnebrog et al. (2019) investigated the response of global integrated water vapor to different forcing agents, but focused on the troposphere.

The goal of this study is to investigate the response of both overworld and LMS SWV to forcing agents with different physical properties. We will explicitly investigate the fast and slow responses in SWV and compare them. We will also investigate how SWV responds to surface temperature change when the climate is forced by different forcing agents.

**2. Method**

**2.1 The PDRMIP set-up**

In this paper, we analyze nine models from the Precipitation Driver and Response Model Intercomparison Project (PDRMIP) (Samset et al., 2016; Myhre et al., 2017; Tang et al., 2018, 2019). These are Coupled Model Inter-comparison Project phase 5 (CMIP5) era models (Table 1) and each performed a baseline and multiple climate perturbation experiments (Table 1).

This subset of the CMIP5 ensemble has a multi-model mean equilibrium climate system (ECS) of 3.6 K, close to the ensemble-average ECS of the entire CMIP5 ensemble (3.3 K) (Zelinka et al. 2020).

In the perturbation experiments, perturbations on a global scale are applied abruptly at the beginning of the model simulation. The five core experiments include a doubling of $CO_2$ concentration (2x$CO_2$), a tripling of $CH_4$ concentration (3x$CH_4$), a 2% increase in solar irradiance (2%Solar), an increase of present-day black carbon concentration or emission by factor of 10 (10xBC), and an increase of present-day $SO_4$ concentration or emission by factor of 5 (5x$SO_4$). In addition to the five core experiments, a subset of models also performed additional perturbation experiments: an increase in CFC-11

concentration from 535 ppt to 5 ppb (hereafter, 10xCFC-11), an increase in CFC-12 concentration from 653.45 ppt to 5 ppb (hereafter, 10xCFC-12), an increase in $N_2O$ concentration from 316 ppb to 1 ppm (hereafter, 3x$N_2O$), an increase troposphericn $O_3$ concentration used in MacIntosh et al. (2016) by factor of 5 (5x$O_3$), and an increase of present-day black carbon with shorter lifetime by factor of 10 (10xBCSLT). We note that indirect chemical effects are not included in the 3x$CH_4$ experiment. Table 1 provides details about the models and the perturbations each one simulated.

The perturbations in GHGs and solar irradiance are relative to the models' baseline simulations, in which the concentration of the GHGs and solar irradiance are either at present-day levels or pre-industrial levels. The perturbations in the aerosols depend on whether it is possible to prescribe aerosol concentrations in the models. For models that are able to prescribe aerosol concentrations, the aerosol perturbations are based on a multi-model mean baseline aerosol concentration in 2000 obtained from the AeroCom Phase II initiative (Myhre et al., 2013). For those that are only able to produce aerosols through

emissions, the perturbation is applied by increasing the emissions by the factors listed above. The 10xBCSLT experiment is performed only by models that are able to prescribe aerosol concentrations.

Each perturbation experiment is performed in two configurations: a fixed sea surface temperatures simulation ("fixed SST") and a fully coupled (slab ocean for CAM4 only) simulation. The fixed SST simulations use the SST climatology at either present-day level or pre-industrial level. The fixed SST simulations are at least 15 years and the coupled simulations are at

least 100 years.

**2.2 Fast response and slow response**

When available, SWV mixing ratio is obtained directly from the specific humidity output by each model simulation. For the models that do not output specific humidity (CAM5, GISS-E2-R, and MIROC-SPRINTARS), we calculate specific humidity by multiplying the models' relative humidity by the saturation mixing ratio with respect to ice calculated using model

temperature and pressure. Responses of specific humidity and relative humidity in the PDRMIP have been investigated by Hodnebrog et al. (2019), but they focused on water vapor in the troposphere.

We define $\Delta$SWV, the change in SWV mixing ratio in response to a particular perturbation, to be the difference between SWV in the perturbed coupled run and that in the baseline coupled run. As discussed above, the $\Delta$SWV can then be broken down into the two components: the fast response ($\Delta$SWV$_{fast}$) and slow response ($\Delta$SWV$_{slow}$). We compute results in the tropical lower stratosphere (70 hPa, 30°N-30°S, hereafter, TLS), in the northern hemispheric (NH) lowermost stratosphere (50°N-90°N at 200 hPa, hereafter, NH LMS), and in the southern hemispheric (SH) lowermost stratosphere (50°S-90°S at 200 hPa, hereafter, SH LMS). Most previous studies have focused on response of water vapor in the TLS (e.g., Gettelman et al., 2010; Shu et al., 2011; Smalley et al., 2017). But recent studies report that the climate is most sensitive to changes in water vapor in the LMS (Solomon et al., 2010; Dessler et al., 2013; Banerjee et al., 2019), so we also investigate that region.

We use the fixed SST simulations to get $\Delta$SWV$_{fast}$, the rapid adjustment in SWV before sea surface temperature changes. $\Delta$SWV$_{fast}$ is the difference between the SWV mixing ratio averaged over the last 10 years in the fixed SST run with the forcing perturbation and the SWV mixing ratio averaged over the last 10 years in the fixed SST baseline simulation. The fixed SST runs have some warming of the land-surface, meaning that our fast response includes a contribution from warming land-surface. We expect this will have a small impact on our results, but it remains one of the uncertainties in our analysis.

We calculate $\Delta$SWV$_{slow}$ as $\Delta$SWV minus $\Delta$SWV$_{fast}$. To estimate the time series of $\Delta$SWV$_{slow}$, we use annual mean $\Delta$SWV over the entire coupled run period (at least 100 years) minus the ten-year average $\Delta$SWV$_{fast}$. To estimate equilibrium $\Delta$SWV$_{slow}$, we use a regression method similar to the methodology introduced by Gregory et al. (2004). The basic concept is that we regress the annual mean global average net downward radiative flux (R) at the top of atmosphere (TOA) against the annual mean $\Delta$SWV averaged at TLS, NH LMS, or SH LMS. The equilibrium $\Delta$SWV is where the linear fit intercepts at R=0. Then we simply subtract $\Delta$SWV$_{fast}$ from the equilibrium $\Delta$SWV to estimate equilibrium $\Delta$SWV$_{slow}$.

These regressions can be very noisy and yield highly uncertain parameters, particularly for perturbations with relatively small amounts of radiative forcing and warming. To account for this, we first fit the R and $\Delta$SWV time series using an exponential function ($y(t) = b + a1 \cdot e^{-t/\tau1} + a2 \cdot e^{-t/\tau2}$), and then do the regression using the fitted time series. For fully coupled models, we constrain $\tau1$ to be within the range of $4\pm2$ years and $\tau2$ to be within the range of $250\pm70$ years; for CAM4, in which the atmosphere is coupled to a slab ocean, we constrain $\tau1$ to be within the range of $4\pm2$ years. We then compute the best fit of all parameters. The ranges for the time constants are based on previous estimations of climate system time scales (Geoffroy et al., 2013). We estimate the $\Delta$SWV-intercept at R=0 by regressing the fitted R and $\Delta$SWV data over the last 30 years, since the relation between R and $\Delta$SWV is not necessarily linear over the entire 100-year period. The slow and fast responses of other variables, such as global average surface temperatures and cold point temperatures are computed using the same method.

We tested this method in a climate model that nearly reaches the equilibrium climate state. We analysed runs of the fully coupled Max Planck Institute Earth System Model version 1.1 (MPI-ESM1.1) (Maher et al., 2019), which has a transient climate response and an effective climate sensitivity near the middle of the CMIP5 ensemble range (Adams and Dessler, 2019; Dessler, 2020). It includes a 2000-year preindustrial control run and a 2614-year abruptly quadrupled $CO_2$ run. The values of $\Delta SWV$ averaged over the last 30 years of the $4xCO_2$ run relative to the control run are 4.60 ppmv in the TLS, 22.40 ppmv in the NH LMS, and 9.69 ppmv in the SH LMS. We expect this to be close to equilibrium $\Delta SWV$ because the trend in global average surface temperature over the last 500 years of the $4xCO_2$ run is 0.02 K per century. We use the regression method to estimate the equilibrium $\Delta SWV$ using MPI-ESM1.1 water vapor mixing ratio time series over the first 100 years and obtain estimates of 4.38 ppmv in the TLS, 20.01 ppmv in the NH LMS, and 9.07 ppmv in the SH LMS; these yield differences of 0.22 ppmv in the TLS, 2.39 ppmv in the NH LMS, and 0.62 ppmv in the SH LMS. Thus, our method underestimates the true equilibrium value by 5% in the TLS, 11% in the NH LMS, and 6% in the SH LMS.

Uncertainty for slow and fast responses of different quantities shown in this paper are obtained from Monte Carlo samples as follows: For each perturbation, we randomly sample with replacement 100,000 times for each model that performed that perturbation and from these samples compute the 2.5%-97.5% percentiles.

## 3. Results

### 3.1 The slow stratospheric water vapor response

We show equilibrium $\Delta SWV_{slow}$ and its percentage contribution to the total equilibrium $\Delta SWV$ in Figure 1. We show results in the TLS (Figs. 1a and 1d), in the NH LMS (Figs. 1b and 1e), and the SH LM (Figs. 1c and 1f). In evaluating the absolute magnitude of $\Delta SWV_{slow}$ in the first column of Fig. 1, we normalize the equilibrium $\Delta SWV_{slow}$ using effective radiative forcing (ERF), so that differences in the magnitude of the forcing do not confound our results.

ERF values used in construction of Fig. 1 are plotted in Fig. 2a; they are calculated as the difference in net radiation at the top of atmosphere (TOA) averaged over the last 10 years between the fixed SST perturbed and baseline simulation. Previous studies have computed the ERF in the PDRMIP using various methods (Richardson et al., 2019; Tang et al., 2019). Our calculation uses the same method as Richardson et al. (2019) "ERF$_{sst}$" and a direct comparison with Richardson et al. (2019) showing good agreement can be found in the supplement (Table S3). The equilibrium global averaged surface temperature changes ($\Delta Ts$), estimated using the regression method described in Section 2.2 and normalized by ERF, are plotted in Fig. 2b. The multi-model mean $\Delta Ts/ERF$ shows general agreement across different perturbations. This quantity is the inverse of the feedback parameter $\lambda$ (e.g. Dessler and Zelinka, 2015), so Fig. 2b implies that the climate sensitivity to these different perturbations is similar, which also agrees with Richardson et al. (2019). We list the ERF and $\Delta Ts$ quantities for each model

and perturbation in Table S1.

In each region, the magnitude of multi-model mean $\Delta SWV_{slow}$/ERF shows general agreement for different perturbations. The magnitudes of $\Delta SWV_{slow}$/ERF in the LMS are larger than those in the TLS (Figs. 1b-c). This is consistent with previous studies, which showed that the long-term trend in SWV over the century in climate models is largest near the LMS tropopause (Dessler et al., 2013; Huang et al., 2016; Banerjee et al., 2019). This reflects different transport pathways into the LMS, including the downward transport by the Brewer-Dobson circulation, quasi-horizontal isentropic mixing from tropical troposphere, and convective influence (Dessler et al., 1995; Holton et al., 1995; Plumb, 2002; Gettelman et al., 2011).

In the LMS, the multi-model mean $\Delta SWV_{slow}$/$\Delta SWV$ ratio is close to 100% for many perturbations (Figs. 1e-f). The latitude band (50º-90º) we choose is somewhat arbitrary, so in the supplement (Fig. S1), we also show $\Delta SWV_{slow}$/ERF and $\Delta SWV_{slow}$/$\Delta SWV$ ratio for water vapor averaged at 200 hPa between 30 and 50 degree latitudes in the NH and SH, respectively, which also show that the $\Delta SWV_{slow}$ plays a dominant role and contributes to close to 100% of the total $\Delta SWV$ for most perturbations. In the TLS, the multi-model mean $\Delta SWV_{slow}$/$\Delta SWV$ ratio is generally above 50%, with a few exceptions. We will discuss this in detail in Section 3.3.

We note that inter-model variability in $\Delta SWV_{slow}$/ERF and $\Delta SWV_{slow}$ is generally consistent for different perturbations. For example, HadGEM3 produces larger responses than the rest of the models for most perturbations (Figs. 1a-c, Table S1). GISS-E2-R and MIROC-SPRINTARS have $\Delta SWV_{slow}$/ERF and $\Delta SWV_{slow}$ values generally below the rest of the models (Figs. 1a-c, Table S1). We have not further investigated the causes of these differences among models; this clearly warrants further investigation.

We also note that CAM5, CanESM2, and MIROC-SPRINTARS produce negative TLS $\Delta SWV_{slow}$/ERF for 10xBC. These negative values are partly contributed by artifacts of the method we use to estimate equilibrium $\Delta SWV_{slow}$, which is the residual of the total equilibrium $\Delta SWV$ minus $\Delta SWV_{fast}$. When differencing two numbers with similar magnitudes, the residual may be quite uncertain. However, the negative values here do not necessarily mean that a BC-induced surface warming results in negative SWV slow response. The direct regression between $\Delta SWV_{slow}$ and surface temperature change described in the next section more accurately describe the relationship for these cases.

**3.2 The slow stratospheric water vapor response and the surface temperature change**

Our results show that, in most climate perturbations analyzed in this study, the equilibrium response of water vapor in both the TLS and the LMS is dominated by $\Delta SWV_{slow}$, which is the component mediated by sea surface temperature change. To directly quantify how SWV responds to surface temperature across a range of different climate change mechanisms, we linearly regress the time series of annual mean $\Delta SWV_{slow}$ over the entire period of the coupled simulations (at least 100

years) against the time series of annual mean global averaged surface temperature change ($\Delta Ts$). We do this regression for each model and perturbation separately. This is similar to the analysis of Banerjee et al. (2019), who did this for quadrupled $CO_2$ perturbation, but we do this for multiple perturbations.

The scatter plot for each perturbation and model is shown in supplement (Figures S3-5). For most perturbations and models, the $\Delta SWV_{slow}$ time series in both the TLS and the LMS is positively correlated with the $\Delta Ts$ time series, supporting the hypothesis that the surface temperature change contributes to the long-term trend in SWV for most cases.

Figure 3 shows the slopes of the regression for all perturbations and models. The corresponding slope values are listed in Table S4. We also list slopes in the unit of %/K in Table S5. The uncertainty of the slopes is obtained from Monte Carlo samples: For each model and perturbation, we first randomly sample the slope 100,000 times, assuming a Gaussian distribution. Then, for each perturbation, we sample from the slope distributions with replacement 100,000 times for each model that performed that perturbation and from these samples compute the multi-model mean and 2.5%-97.5% percentiles.

In both the TLS and LMS, the slopes from different perturbations show general agreement (Fig. 3); this is also true for water vapor averaged at 200 hPa between 30 and 50 degree latitudes in the NH and SH (Fig. S2). In the TLS, the multi-model and multi-perturbation average slope is 0.35 ppmv $K^{-1}$ with a 95% confidence interval of 0.28-0.44 ppmv $K^{-1}$ (Fig. 3a). The LMS $\Delta SWV_{slow}$ time series has stronger correlations with the $\Delta Ts$ time series (Figures S3-5) and produces larger sensitivities (Figs. 3b-c). Specifically, the multi-model and multi-perturbation mean slope is 2.1 ppmv $K^{-1}$ in the NH, and is 0.97 ppmv $K^{-1}$ in the SH, with 95% confidence intervals of 1.82-2.39 ppmv $K^{-1}$ and 0.79-1.15 ppmv $K^{-1}$, respectively. Our results are similar to those of Dessler et al. (2013) and Smalley et al. (2017) despite the fact that they used 500-hPa temperature as their regressor.

We show that the relation between $\Delta SWV_{slow}$ and $\Delta Ts$ time series can be extended to the entire stratosphere (Figs. 4a). We re-gridded the zonal mean $\Delta SWV_{slow}$ from all models and perturbations onto the same pressure-latitude grid (10 hPa above 100 hPa and 50 hPa below 100 hPa, 4 degrees latitude) and regress the $\Delta SWV_{slow}$ time series at each grid point against global average $\Delta Ts$ time series. The multi-model and multi-perturbation average slope of the linear fit at each grid point is shown in Fig. 4a (Figures for each individual perturbation are shown in Fig. S6). Since the vertical gradient of water vapor is large, we plot the percentage change of mixing ratio per degree K relative to the baseline. Lapse rate tropopause, the lowest level where the lapse rate decreases to 2 K $km^{-1}$, also plotted, is obtained using the atmospheric temperatures from the baseline coupled run and multi-model mean.

We clearly see the larger sensitivity of $\Delta SWV_{slow}$ to $\Delta Ts$ in the LMS than in the overworld. In the LMS, the slope has a hemispheric asymmetry, with larger values in the NH. This is consistent with previous studies, which showed that isentropic transport brings more tropospheric water vapor to the NH than the SH (Pan et al., 1997, 2000; Dethof et al., 1999, 2000;

Ploeger et al., 2013). In addition, convective moistening may be more important to the NH due to more land in the NH and, consequently, more convection (Dessler and Sherwood, 2004; Smith et al., 2017; Ueyama et al., 2018; Wang et al., 2019). We also see large responses in the tropical upper troposphere, which is the main part of the tropospheric water vapor feedback. The sensitivity declines as one ascends through the TTL. Once above the TTL, the sensitivity in the overworld is relatively uniform with altitude.

## 3.3 The fast stratospheric water vapor response

Figure 1 also shows the $\Delta SWV_{fast}$ normalized by the ERF (Figs. 1g-i) as well as its contribution to total equilibrium $\Delta SWV$ (Figs. 1j-l). As discussed previously, $\Delta SWV_{fast}$ is the rapid adjustment in SWV, before the sea surface temperatures respond. For most perturbations, especially in the LMS, $\Delta SWV_{fast}/ERF$ is smaller than $\Delta SWV_{slow}/ERF$, with a magnitude of a few tenths of a ppmv$\cdot$(Wm$^{-2}$)$^{-1}$.

For 2xCO$_2$, the near-zero TLS $\Delta SWVf_{ast}/ERF$ is the result of cancellation between cooling by a strengthening Brewer-Dobson circulation and increased local radiative heating (Lin et al., 2017). Some other GHG forcing agents, however, produce larger TLS $\Delta SWV_{fast}/ERF$ and contributions in the TLS. The multi-model mean $\Delta SWV_{fast}$ from 10xCFC-12 and 10xCFC-11 contribute about half of the total $\Delta SWV$, respectively (Fig. 1j). This is a consequence of halocarbons producing more TTL warming per Wm$^{-2}$ by efficiently absorbing upwelling longwave radiation from the troposphere in the atmospheric window (Forster et al., 1997; Jain et al., 2000; Forster and Joshi, 2005). Fig. 5 shows the fast temperature response per unit ERF due to different perturbations and it shows heating in the TTL for both 10xCFC-12 and 10xCFC-11.

The 3xCH$_4$ also includes some models that produce large TLS $\Delta SWV_{fast}/ERF$ magnitudes. This is likely due to TTL heating (Fig. 5) by CH$_4$ shortwave absorption, which is explicitly treated in some models, including CAM5, CanESM2, MPI-ESM, and MIROC-SPRINTARS (Smith et al., 2018). These models are also the ones that produce the largest TLS $\Delta SWV_{fast}$ contributions (Figs. 1g and 1j).

Increases of tropospheric O$_3$ (in the 5xO$_3$ experiment) reduce the upwelling longwave radiation, which cools the stratosphere (Ramaswamy and Bowen, 1994; Berntsen et al., 1997; Forster et al., 1997). The longwave radiation absorbed heats the TTL region (Fig. 5), resulting in larger TLS $\Delta SWV_{fast}/ERF$ magnitude than $\Delta SWV_{slow}/ERF$ and larger contributions to total equilibrium $\Delta SWV$ (77%) (Figs. 1g and 1j). There is also heating in the LMS, resulting in larger LMS $\Delta SWV_{fast}/ERF$ magnitude than $\Delta SWV_{slow}/ERF$ (Figs. 1h-j). We note that our conclusion on 5xO$_3$ is based on only one model, MIROC-SPRINTARS.

$\Delta SWV_{fast}$ from 10xBC dominates total equilibrium $\Delta SWV$ in the TLS, with multi-model mean contribution of 84%. The magnitude of the multi-model mean $\Delta SWV_{fast}/ERF$ from 10xBC is also larger than any other perturbations in each region.

This occurs because the 10xBC strongly absorbs shortwave radiation, causing large heating of the tropopause region in both the tropics and extra-tropics. Figure 5 shows the 10xBC gives by far the most warming per unit ERF, which is consistent with the vertical profile of fast temperature response shown in Stjern et al. (2017).

The 10xBC $\Delta SWV_{fast}$/ERF in the NH and SH LMS contributes to about 50% of the total equilibrium $\Delta SWV$, with smaller magnitudes in the SH (Figs. 1h-i and 1k-l). This is because the total amount of black carbon is smaller in the SH (Myhre et

al., 2017), since black carbon is a combustion product and is predominantly emitted over the NH continents (Ramanathan and Carmichael, 2008). The 10xBCSLT $\Delta SWV_{fast}$ also contributes about 50% of the total 10xBCSLT $\Delta SWV$. The 10xBCSLT does not produce as strong a $\Delta SWV_{fast}$/ERF as 10xBC, since the reduction in BC lifetime leads to less BC in the TTL and therefore less heating per unit ERF.

We quantify control of TLS $\Delta SWV_{fast}$ by the fast TTL temperature adjustments across a range of different climate

perturbations by regressing the TLS $\Delta SWV_{fast}$ against the fast response of the cold point temperature ($\Delta TCP_{fast}$). To estimate $\Delta TCP_{fast}$ in the models, we first find the minimum temperature in the profile at each grid point in the fixed SST runs (no interpolation is done, we simply find the minimum temperature on the output model levels). These minimum temperatures are then averaged between 30°N – 30°S to yield $TCP_{fast}$ in each run. $\Delta TCP_{fast}$ is the difference between $TCP_{fast}$ in the perturbed model run minus that in the baseline runs.

We find that TLS $\Delta SWV_{fast}$ is strongly correlated with $\Delta TCP_{fast}$ across all perturbations and models (Fig. 6a), with a slope of 0.52 ppmv K$^{-1}$ and a 95% confidence interval of 0.43 to 0.61 ppmv K$^{-1}$. Randel and Park (2019) pointed out that the slope from the Clausius-Clapeyron relationship evaluated near the tropical tropopause is close to this value, about 0.5 ppmv K$^{-1}$. We also tested the relationship between TLS $\Delta SWV_{slow}$ and slow response of the cold point temperature ($\Delta TCP_{slow}$) across all perturbations and models, yielding a slope of 0.72 ppmv K$^{-1}$. However, for the slow response, correlation does not

necessarily prove causality, since Dessler et al. (2016) showed that, in two climate models at least, a significant fraction of the long-term trend was due to increases in convective moistening, which bypasses the TTL cool trap. Therefore this relationship for the slow response could arise from either TCP control or a process that correlates with it, such as deep convective injection of ice, or some combination.

We also separately plot the slopes between $\Delta SWV_{fast}$ and $\Delta TCP_{fast}$ for each perturbation (Figs. 6d-f). For the perturbations

that have more than five participating models, including 2xCO$_2$, 3xCH$_4$, 2%Solar, 10xBC, 5xSO$_4$, and 10xCFC-12, we calculate the linear regression between $\Delta SWV_{fast}$ and $\Delta TCP_{fast}$ from the models and show the slopes and 95% confidence intervals. For the perturbations that have fewer participating models, including 10xCFC11, 3xN$_2$O, 5xO$_3$, and 10xBCSLT, we plot the ratio $\Delta SWV_{fast}$/$\Delta TCP_{fast}$ and show only the multi-model mean. The slopes produced by different perturbations show general agreement (Fig. 6d). The larger uncertainty in the slopes produced by 2%Solar and 10xCFC-12 occurs because

both the $\Delta TCP_{fast}$ and $\Delta SWV_{fast}$ produced by different models are similar and therefore the slope of the linear regression is

uncertain. Overall, we find that the fast response of TTL temperature is a good predictor for the TLS $\Delta SWV_{fast}$ across a range of different climate mechanisms and across multiple models.

For the LMS $\Delta SWV_{fast}$, the $\Delta TCP_{fast}$ does not show a control as strong as that in the TLS (Figs. 6b-c) due to the fact that TTL temperatures are only one factor that influences the LMS. In addition, the regression between $\Delta SWV_{fast}$ and $\Delta TCP_{fast}$
across all perturbations at each grid point in the pressure-latitude domain shows that the slope (% $K^{-1}$) follows the transport pattern of the BDC (Fig. 4b). The slope is large in the tropical overworld stratosphere and become weaker as one moves poleward and downward in the extra-tropics below 150 hPa. The value is lower in the LMS, again consistent with the fact that water vapor in the LMS is controlled by several processes, not just TTL cold-point temperature. Clearly, more work on this is warranted.

**4. Historical changes in SWV**

Given the importance of SWV change, we now ask whether our results can help us understand historical variations in TLS $\Delta SWV$ over 1980-2010 (Figure 7). To do this, we estimate historical values of $\Delta SWV_{slow}$ and $\Delta SWV_{fast}$ based on the PDRMIP results, historical surface temperature change, and historical radiative forcing. For the slow component (blue in Fig. 7a), we multiply 0.35 ppmv $K^{-1}$, the multi-model multi-perturbation mean sensitivity of the PDRMIP TLS $\Delta SWV_{slow}$ to
$\Delta Ts$, by the historical surface temperature change over 1980-2010. For the fast component (orange in Fig. 7a), we multiply the multi-model mean PDRMIP TLS $\Delta SWV_{fast}$/ERF value for each perturbation by the corresponding historical radiative forcing and then sum it up. We also show the fast component of the historical $\Delta SWV$ contributed by each historical forcing agent in Fig. 7b. This is similar to the analysis done by Hodnebrog et al. (2019) in their Figure 6, where they used this method to estimate the historical water vapor lifetime change based on the PDRMIP results.

The historical surface temperature change and radiative forcing data used in this analysis are listed in Table 2. The historical radiative forcing we use here is defined as the change in net downward radiative flux at the tropopause, after adjustments in the stratospheric temperatures, while the surface and troposphere are held unperturbed (Myhre et al. 2013). This is different from the ERF we use in the PDRMIP calculations, which introduced uncertainties in the fast component of the historical $\Delta SWV$ we estimate based on PDRMIP.

Figure 7a shows our estimate that climate change over 1980-2010 has increased TLS SWV by 0.51±0.16 ppmv (Fig. 7a). 36% is due to the slow component, although this is probably an overestimate because our sensitivity value estimated using the PDRMIP results are for long-term. We find the rest of the $\Delta SWV$, 64%, is due to the fast component, mainly from black carbon. We have also calculated the SWV sensitivity and SWV fast response over 35°N-45°N between 100-80 hPa to re-compute the historical 1980-2010 $\Delta SWV$ using the same method, which is 0.65±0.20 ppmv. This value shows reasonable
agreement with the SWV increase measured by Hurst et al. (2011) of 0.71±0.26 ppmv over Boulder between 16-18 km over

1980-2010.

Dessler et al. (2014) and Hegglin et al. (2014) argue that there is not a detectable trend over this period. Such a conclusion is not inconsistent with ours because any actual trend estimate has to contend with short-term interannual variability (i.e, like that from the QBO and Brewer-Dobson Circulation variability), which can mask a small trend. Our estimate of the trend is
based on sensitivity estimated from 100 year-run and therefore short-term interannual variability has a small impact. Given continuous reliable long-term SWV observation record in the future, one will be able to better test the model-predicted values.

For the fast component of the estimated historical $\Delta$SWV, radiative forcing by BC plays the dominant role (Fig. 7b). Uncertainties exist in the historical BC radiative forcing we use in this analysis, which is shown in the IPCC AR5 (Myhre et
al. 2013). In addition, Allen et al. (2019) pointed out that the radiative effect by BC in the PDRMIP is different from that shown in models using observationally constrained aerosol forcing, which may overestimate the heating in the UTLS region. However, Allen et al. (2019) also noted that uncertainties exist in their observationally constrained aerosol forcing. The uncertainties in the impact of BC forcing on SWV clearly merit more analysis in the future.

## 5. Conclusions

It is of great interest for the climate community to understand how SWV changes when the climate changes since SWV plays an important role in the Earth's radiative budget and stratospheric ozone chemistry (Solomon et al., 1986, 2010; Dvortsov and Solomon, 2001; Forster and Shine, 2002). In this study, we investigate the response of stratospheric water vapor (SWV) to a range of different climate forcing mechanisms using a multi-model and multiple forcing agent framework. We use output from nine CMIP5 models participating the PDRMIP. Each model performs a baseline and up to 10 climate
perturbation experiments, including $2\times CO_2$, $3\times CH_4$, 2%Solar, 10xBC, $5\times SO_4$, 10xCFC-11, 10xCFC-12, $3\times N_2O$, $5\times O_3$, and 10xBCSLT (Table 1). Each perturbation is performed in two configurations, including fixed SST simulations (at least 15 years) and fully coupled simulations (at least 100 years).

To better understand the SWV response ($\Delta$SWV), we partition it into two parts: the slow response ($\Delta$SWV$_{slow}$) and the fast response ($\Delta$SWV$_{fast}$). The $\Delta$SWV$_{fast}$ is the change in response to a perturbation on short time scales, before the surface temperature has responded.
temperature has responded. $\Delta$SWV$_{slow}$ occurs on longer time scales and is coupled to the surface temperature change. Our results show that, for most perturbations, $\Delta$SWV in the tropical lower stratosphere (TLS) and in the lowermost stratosphere (LMS) (200 hPa, 50°N-90°N and 50°S-90°S) is dominated by $\Delta$SWV$_{slow}$ (Fig. 1).

Analysis of $\Delta$SWV$_{slow}$ shows that a warming surface increases SWV (Figures S3-5). Furthermore, the response of SWV to the surface temperature change has a similar sensitivity across different climate perturbations in both the overworld

stratosphere and the lowermost stratosphere (Figs. 3 and 4a). Specifically, the multi-model and multi-perturbation mean slope is 0.35 ppmv $K^{-1}$ in the TLS, 2.1 ppmv $K^{-1}$ in the northern hemispheric (NH) LMS, and 0.97 ppmv $K^{-1}$ in the southern hemispheric (SH) LMS (Fig. 3).

$\Delta SWV_{slow}$ in the LMS is more sensitive to $\Delta Ts$ than the tropical overworld, reflecting different transport pathways into the LMS compared to the overworld (Dessler et al., 1995; Holton et al., 1995; Plumb, 2002; Gettelman et al., 2011). The $\Delta SWV_{slow}$ in the NH LMS is more sensitive than the SH LMS, consistent with hemispheric asymmetries in the isentropic transport and convective moistening reported by previous studies (Pan et al., 1997, 2000; Dethof et al., 1999, 2000; Dessler and Sherwood, 2004; Ploeger et al., 2013; Smith et al., 2017; Ueyama et al., 2018; Wang et al., 2019).

The fast response of SWV from most perturbations are weak compared to the slow response and therefore plays a smaller role in $\Delta SWV$ (Fig. 1). In the TLS, for forcing agents that directly heat tropopause levels (Fig. 5), $\Delta SWV_{fast}$ makes a larger contribution to $\Delta SWV$. In particular, when climate is perturbed by 10xBC, the $\Delta SWV_{fast}$ dominates the $\Delta SWV_{slow}$ and has larger magnitude than any other perturbed simulations. This occurs because black carbon absorbs shortwave radiation in the atmosphere and directly heats the temperatures at tropopause levels. Other forcing agents also heat the tropopause levels and increase $\Delta SWV_{fast}$ through absorption of shortwave radiation or longwave radiation at the atmospheric window range (3xCH$_4$, 5xO$_3$, 10xBCSLT, 10xCFC-12, 10xCFC-11), but are not as strong as 10xBC.

The TLS $\Delta SWV_{fast}$ is controlled by the fast response of the cold point temperature across different climate change mechanisms (Fig. 6), with a slope of 0.52 ppmv $K^{-1}$, which is consistent with the Clausius-Clapeyron relationship evaluated near the tropical tropopause (Randel and Park, 2019). The control of cold point temperature fast response over $\Delta SWV_{fast}$ is stronger in the tropical overworld and becomes weaker at higher latitudes and altitudes below 150 hPa in the LMS (Fig. 4b).

*Data availability:* The PDRMIP data can be downloaded from this website: https://cicero.oslo.no/en/PDRMIP.

*Competing interests.* The authors declare that they have no conflict of interest.

*Author contribution:* Xun Wang performed analyses and wrote the paper. Andrew E. Dessler provided the conceptualization, guidance, and editing.

*Acknowledgments:* This work was supported by NASA grants 80NSSC18K0134 and 80NSSC19K0757. This work was also supported by the National Center for Atmospheric Research, which is a major facility sponsored by the National Science Foundation under Cooperative Agreement No. 1852977. Any opinions, findings and conclusions or recommendations

expressed in this material do not necessarily reflect the views of the National Science Foundation. We would like to acknowledge the PDRMIP modelling groups and helpful discussions with Andrew Gettelman and William Randel.

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

**Table 1: Columns 1-6: Description of PDRMIP models (Myhre et al., 2017). © American Meteorological Society. Used with permission. Column 7: List of perturbation experiments used in this study.**

| Model | Version | Resolution | Ocean setup | Aerosol setup | Key references | Perturbation experiments |
|---|---|---|---|---|---|---|
| Second Generation Canadian Earth System Model (CanESM2) | 2010 | 2.8°×2.8°, 35 levels | Coupled ocean | Emissions | (Arora et al., 2011) | $2xCO_2$, $3xCH_4$, 2%Solar, 10xBC, $5xSO_4$ |
| Community Earth System Model, version 1 (Community Atmosphere Model, version 4) [CESM1(CAM4)] | 1.0.3 | 2.5°×1.9°, 26 levels | Slab ocean | Fixed concentrations | (Neale et al., 2010; Gent et al., 2011) | $2xCO_2$, $3xCH_4$, 2%Solar, 10xBC, $5xSO_4$, 10xCFC-12, $3xN_2O$, 10xBCSLT |
| CESM1 CAM5 | 1.1.2 | 2.5°×1.9°, 30 levels | Coupled ocean | Emissions | (Hurrell et al., 2013; Kay et al., 2015; Otto-Bliesner et al., 2016) | $2xCO_2$, $3xCH_4$, 2%Solar, 10xBC, $5xSO_4$, 10xCFC-12 |
| Goddard Institute for Space Studies Model E2, coupled with the Russell ocean model (GISS-E2-R) | E2-R | 2°×2.5°, 40 levels | Coupled ocean | Fixed concentrations | (Schmidt et al., 2014) | $2xCO_2$, $3xCH_4$, 2%Solar, 10xBC, $5xSO_4$, 10xCFC-12, 10xBCSLT |
| Hadley Centre Global Environment Model, version 2—Earth System (includes Carbon Cycle configuration with chemistry) (HadGEM2-ES) | 6.6.3 | 1.875°×1.25°, 38 levels | Coupled ocean | Emissions | (Collins et al., 2011; Martin et al., 2011) | $2xCO_2$, $3xCH_4$, 2%Solar, 10xBC, $5xSO_4$, 10xCFC-12, 10xCFC-11, |

| | | | | | | $3xN_2O$ |
|---|---|---|---|---|---|---|
| HadGEM3 | Global Atmosphere 4.0 | $1.875°\times1.25°$, 85 levels | Coupled ocean | Fixed concentrations | (Bellouin et al., 2011; Walters et al., 2014) | $2xCO_2$, $3xCH_4$, 2%Solar, 10xBC, $5xSO_4$, 10xCFC-12 |
| L'Institut Pierre-Simon Laplace Coupled Model, version 5A (IPSL-CM5A) | CMIP5 | $3.75°\times1.875°$, 39 levels | Coupled ocean | Fixed concentrations | (Dufresne et al., 2013) | $2xCO_2$, $3xCH_4$, 2%Solar, 10xBC, $5xSO_4$ |
| Max Planck Institute Earth System Model (MPI-ESM) | 1.1.00p2 | T63, 47 levels | Coupled ocean | Climatology, year 2000 | (Giorgetta et al., 2013) | $2xCO_2$, $3xCH_4$, 2%Solar |
| Model for Interdisciplinary Research on Climate-Spectral Radiation-Transport Model for Aerosol Species (MIROC-SPRINTARS) | 5.9.0 | T85 (approx. $1.4°\times1.4°$), 40 levels | Coupled ocean | Hemispheric Transport Air Pollution, phase 2 Emissions | (Takemura, 2005; Takemura et al., 2009; Watanabe et al., 2010) | $2xCO_2$, $3xCH_4$, 2%Solar, 10xBC, $5xSO_4$, 10xCFC-12, 10xCFC-11, $3xN_2O$, $5xO_3$ |

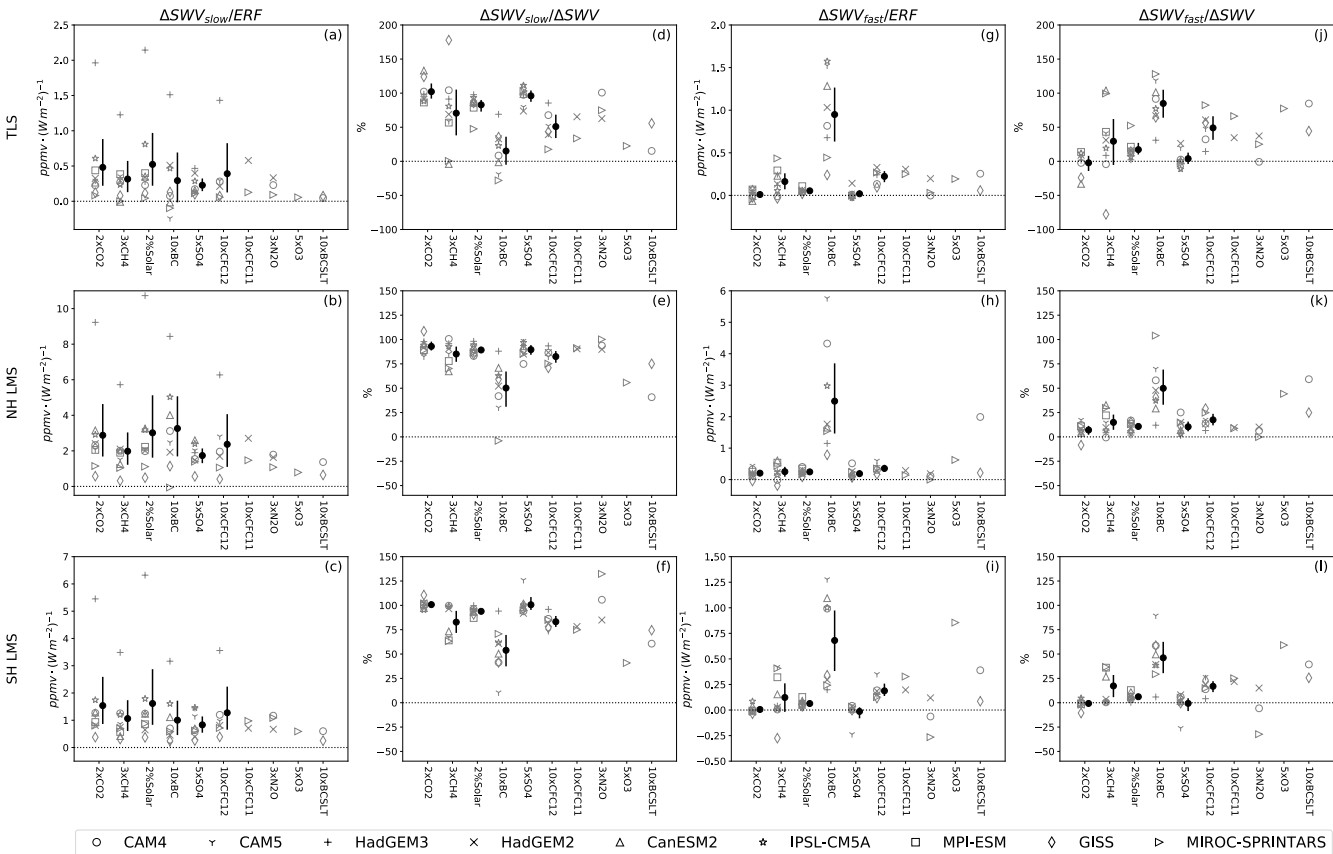


**Figure 1. Panels (a)-(c): equilibrium $\Delta SWV_{slow}$ normalized by ERF (ppmv·(Wm$^{-2}$)$^{-1}$) in TLS (70 hPa, 30°N-30°S), NH LMS (200 hPa, 50°N-90°N), and SH LMS (200 hPa, 50°S-90°S). Panels (d)-(f): Contribution (%) of equilibrium $\Delta SWV_{slow}$ to total equilibrium $\Delta SWV$. Panels (g)-(i): $\Delta SWV_{fast}$ normalized by ERF (ppmv·(Wm$^{-2}$)$^{-1}$). Panels (j)-(l): Contribution (%) of $\Delta SWV_{fast}$ to**

**total equilibrium $\Delta SWV$. The marker shapes indicate results from different models. For perturbations that are performed by more than three models, the solid circles and error bars for each perturbation plotted in weighted black are multi-model mean and 2.5%-97.5% percentiles of the model samples. Note that in the second and fourth columns, we took out models with extremely small $\Delta SWV$ magnitudes that yield extremely large $\Delta SWV_{slow}/\Delta SWV$ and $\Delta SWV_{fast}/\Delta SWV$ ratios.**

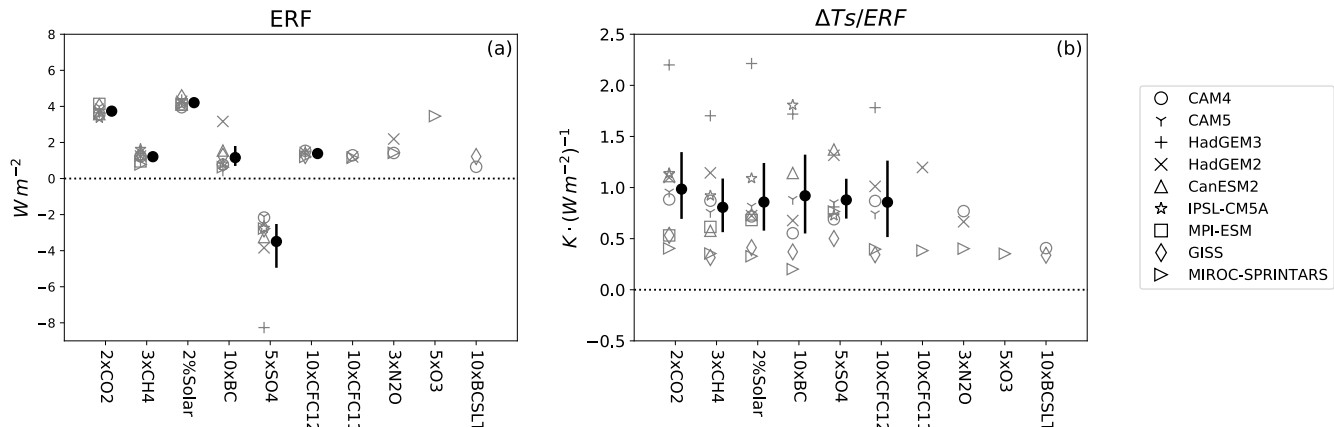

**Figure 2. Panel (a): Global average ERF (W m⁻²) at the top of atmosphere. Panel (b): Global averaged surface temperature change per unit ERF (K·(W m⁻²)⁻¹). The marker shapes indicate results from different models. For perturbations that are performed by more than three models, the solid circles and error bars for each perturbation plotted in weighted black are multi-model mean and 2.5%-97.5% percentiles of the model samples.**

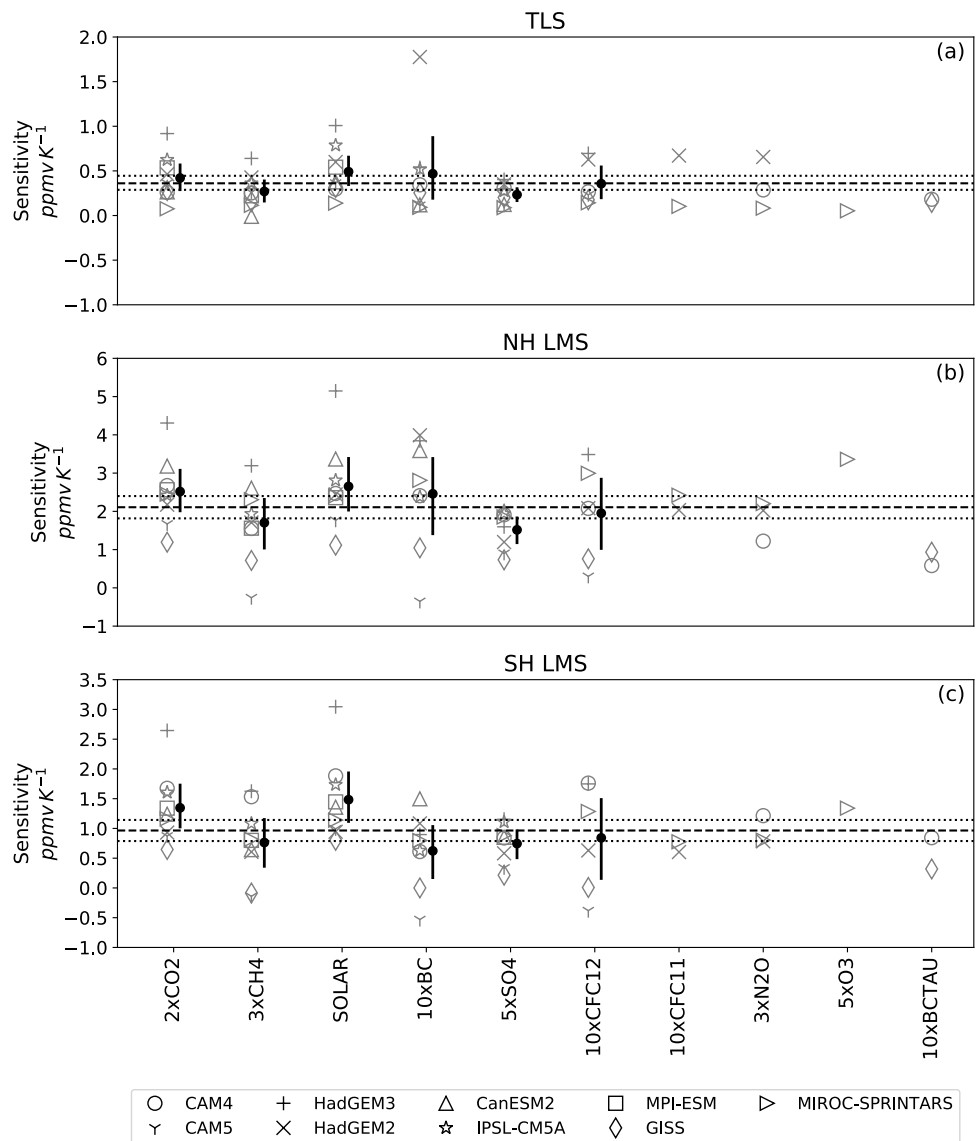


**Figure 3. Slopes (ppmv K⁻¹) from the linear regression between annual mean ΔSWV$_{slow}$ time series and annual mean ΔTs time series. The marker shapes indicate results from different models. For perturbations that are performed by more than three models, the solid circles and error bars for each perturbation plotted in weighted black are multi-model mean and 2.5%-97.5% percentiles of the model samples. The horizontal dashed line is the multi-model mean of all slopes, and the horizontal dotted lines**

**are 2.5%-97.5% percentiles of the model samples.**

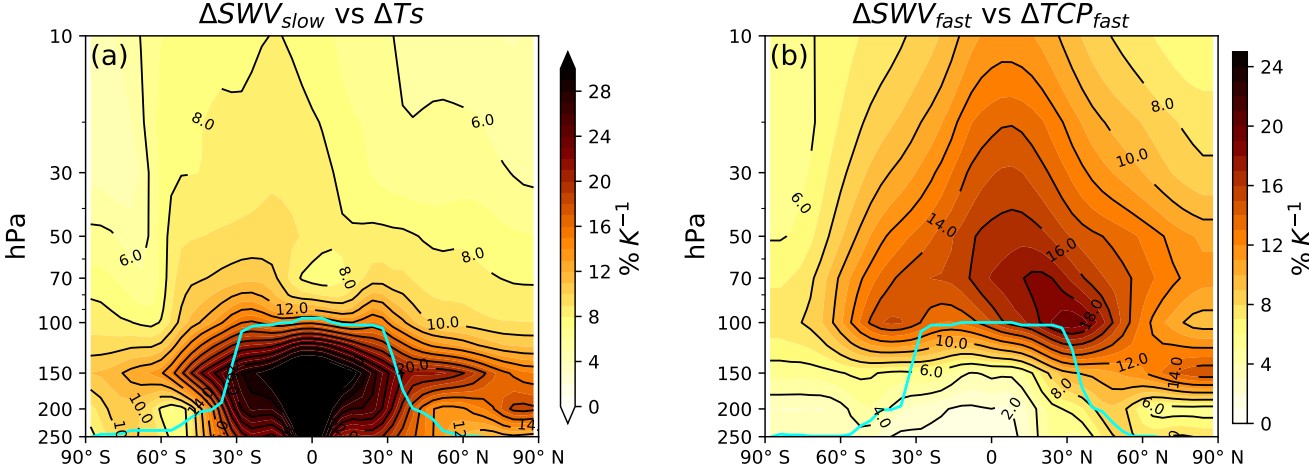

**Figure 4.** Panel (a): Multi-model and multi-perturbation mean slope (% K$^{-1}$) from the regression between annual mean time series of $\Delta SWV_{slow}$ at each latitude grid point and pressure level and annual mean time series of global average $\Delta Ts$. Panel (b): Slope (% K$^{-1}$) from the regression between $\Delta SWV_{fast}$ (ppmv) at each latitude grid point and pressure level and $\Delta TCP_{fast}$ (K). The solid cyan line is the multi-model mean lapse rate tropopause derived from the baseline simulations.

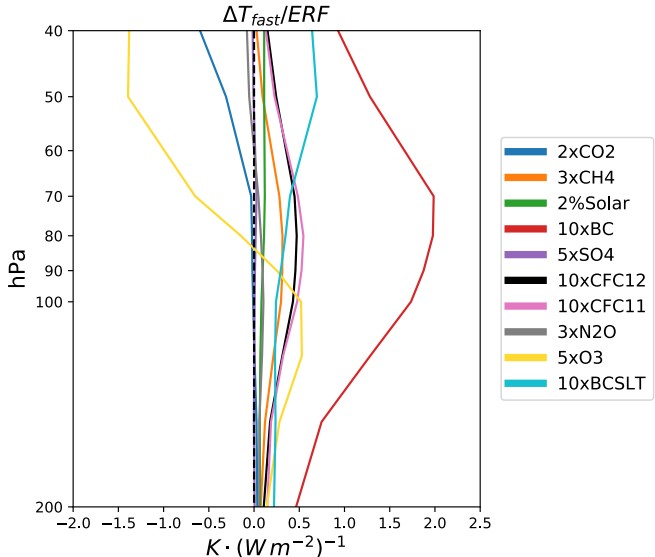

**Figure 5.** Profiles of fast temperature response normalized by ERF (K·(Wm$^{-2}$)$^{-1}$) between 200 and 40 hPa, and averaged over 30°N-30°S. The color coding indicates results from different perturbations. Each profile is the multi-model mean.

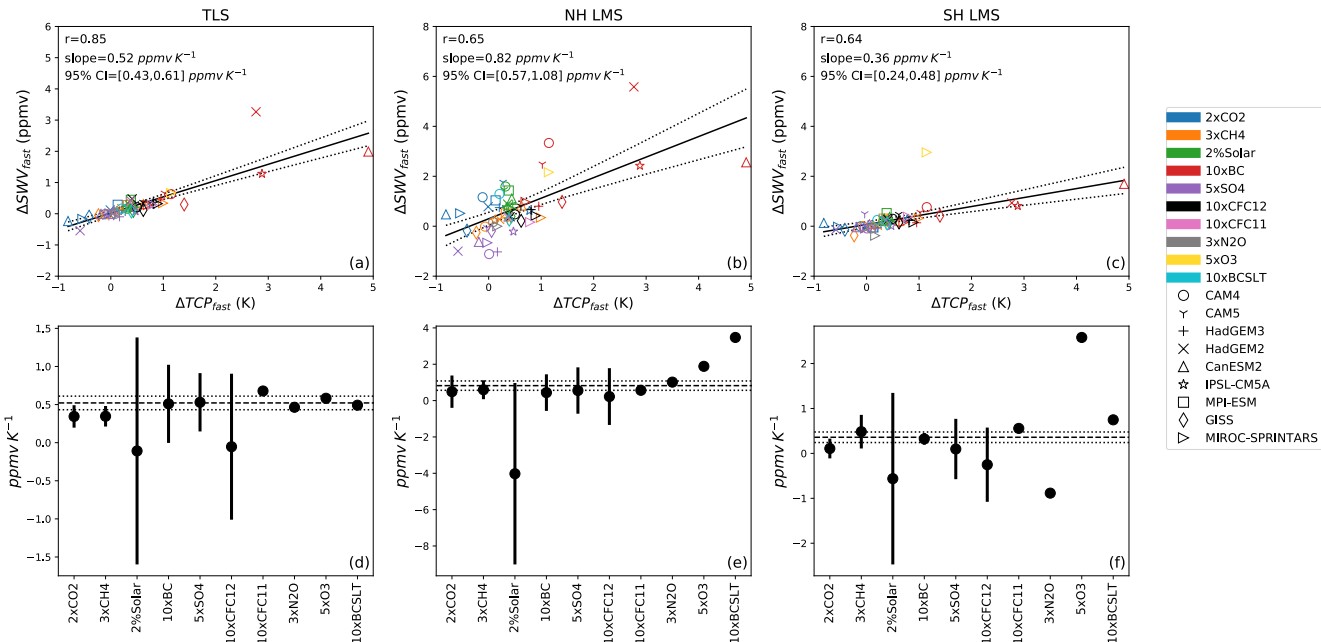

**Figure 6.** Panels (a)-(c): Linear regression between ΔSWV$_{fast}$ (ppmv) and ΔTCP$_{fast}$ (K) from all models and perturbations. The color coding indicates different perturbations, while the marker shapes indicate results from different models. The black solid line is the linear fit of the regression. The black dotted lines indicate the linear fits within the 95% confidence interval, estimated using a t-test. Panels (d)-(f): Slopes and their 95% confidence intervals (for perturbations that are performed by more than three models) obtained from linear regression between ΔSWV$_{fast}$ (ppmv) and ΔTCP$_{fast}$ (K) for each individual perturbation. The black dashed lines and dotted lines are the slopes and their 95% confidence intervals of regressions in (a)-(c).

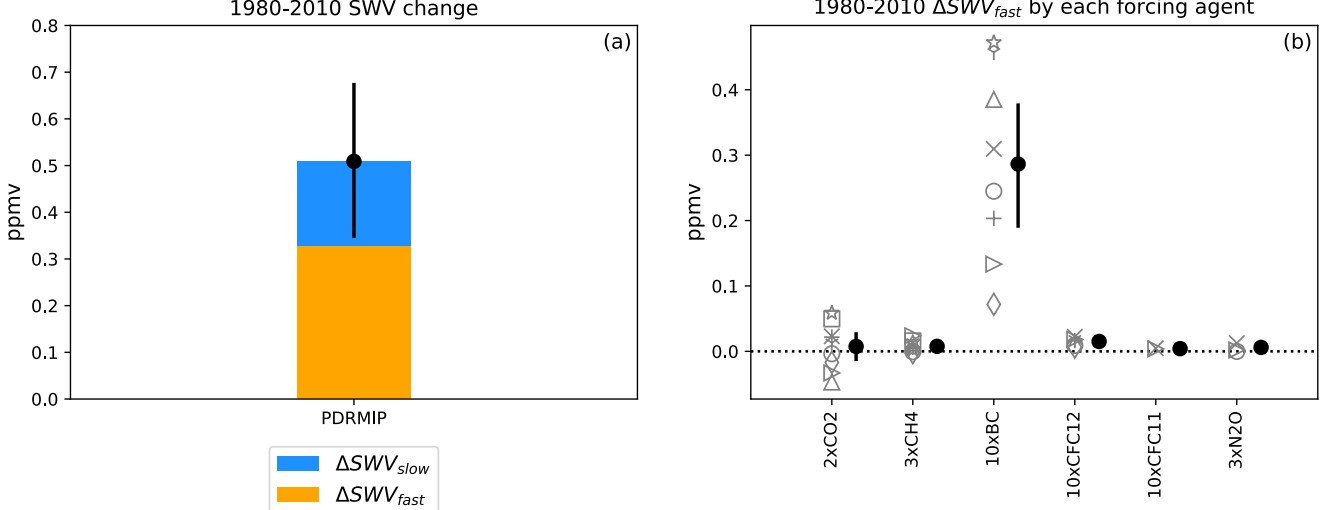

**Figure 7. (a) TLS (30°S-30°N, 70 hPa) SWV change over 1980-2010 estimated using PDRMIP results. Blue indicates the component contributed by the slow response, while orange indicates the component contributed by the fast response. (b) The fast component of the PDRMIP-estimated 1980-2010 SWV change contributed by each historical forcing agent. The solid circles are multi-model mean. The error bars are 2.5%-97.5% percentiles of the model samples; in Panel (b) they are shown for perturbations that are performed by more than three models.**

`

**Table 2: Historical global average surface temperature change and radiative forcing (RF) by Greenhouse gases (GHGs) and halocarbons over 1980-2010. The SWV change over 1980-2010 estimated using PDRMIP results is also listed, including the total SWV change, the slow component, and the fast component. For the fast component of SWV change contributed by each forcing agent, multi-model mean results are listed. The uncertainties are 2.5%-97.5% percentiles of the model samples.**

| | |
|---|---|
| GMST[a] (K) | 0.506 |
| Total $\Delta$SWV (ppmv) | 0.51±0.16 |
| $\Delta$SWV$_{slow}$ (ppmv) | 0.18±0.04 |
| $\Delta$SWV$_{fast}$ (ppmv) | 0.32±0.12 |

| Forcing agents | RF (Wm$^{-2}$) | $\Delta$SWV$_{fast}$ by each forcing agent (ppmv) |
|---|---|---|
| CO$_2$[b] | 0.715 | 0.007±0.022 |
| CH$_4$[c] | 0.055 | 0.008±0.005 |
| BC[d] | 0.3 | 0.286±0.095 |
| CFC-12[e] | 0.068 | 0.015±0.004 |
| CFC-11[f] | 0.015 | 0.004 |
| N$_2$O[g] | 0.042 | 0.005 |

**a:** We used NOAA Merged Land Ocean Global Surface Temperature Analysis V5 (Zhang et al. 2020) to compute the global surface temperature change. We use values averaged over 2005-2015 minus that averaged over 1975-1985.

**b, c, e, f, and g:** We compute the RFs using the formulae listed in Table 3 of Myhre et al. (1998). These formulae were also used to compute RFs of CO$_2$, CH$_4$, and N$_2$O in IPCC reports (Myhre et al. 2013).

**b, c, and g:** Concentrations of GHGs were used to compute RFs. CO$_2$ and CH$_4$ are samples collected in glass flasks at Cold Bay, Alaska, United States (CBA) from the ERSL GML website (Dlugokencky et al. 2020). N$_2$O is from the Combined Nitrous Oxide data from the NOAA/ESRL Global Monitoring Division. For CO$_2$, concentrations averaged over 2005-2015

and averaged over 1978-1985 are used. For $CH_4$, concentrations averaged over 2005-2015 and averaged over 1983-1985 are used. For $N_2O$, concentration averaged over 2005-2015 and averaged over 1977-1985 are used.

**e-f:** Concentrations of CFC-12 and CFC-11 were used to compute RFs. We use CFC-12 and CFC-11 data from combined stations from the NOAA/ESRL Global Monitoring Division. Concentrations averaged over 2005-2015 and averaged over 1977-1985 are used.

**d:** We use 0.4 $Wm^{-2}$, the BC RF between 1750-2011 reported in IPCC AR5, minus 0.1 $Wm^{-2}$, the BC RF between 1750-1993 reported in 1995 IPCC report (See Table 8.4 of Myhre et al. 2013).