# Peer review of "The response of stratospheric water vapor to climate change driven by different forcing agents"

_Atmospheric Chemistry and Physics, 2020_

## Referee Comment (RC1) · Anonymous Referee #2 · 15 Jul 2020

This is a useful and mostly clearly written analysis of the behavior of stratospheric water vapor (SWV) in a set of idealised multi-model (PDRMIP) climate simulations. Subject to satisfactory modification it should become acceptable for publication. Some of the suggestions below are discetionary, in my view, but I feel acting on them would lead to a more valuable paper.

GENERAL COMMENTS

1. The paper largely focuses on interpreting the multi-model responses. While this is of course useful, it stops short of relating the new understanding to any real-world changes in SWV. Indeed, a reader could incorrectly infer (by omission) from the abstract and conclusions that BC is the dominant (fast) driver of SWV changes, as no mention is made of the idealised nature or size of the perturbations.

I feel this is a significant weakness and I encourage more discussion. How much does this work help in understanding past and possible future SWV changes? Is it possible (as has been done in some of the PDRMIP papers such as Samset et al. 10.1038/s41467-018-04307-4 and the Hodnebrog paper referred to below) to make a suitable composite of the individual perturbations to derive responses that have or may in future occur? This would require some additional critical assessment of the relevance of the PDRMIP perturbations to the real world. In particular, note there has been some discussion of how the PDRMIP BC perturbations compare to observations (Allan et al, https://doi.org/10.1038/s41612-019-0073-9). I see that Allan et al's supplementary figure S5 shows the impact of BC on UTLS temperature changes in the PDRMIP models, and that needs mentioning here, in the context of the BC conclusions in this paper. Similarly, the CFC perturbations are much higher than any present or likely future changes.

2. As noted in the specific comments, I feel that there is inadequate recognition that some of the results presented here are also presented, either explicitly or implicitly, in some earlier papers from the PDRMIP group – this is particularly so for the ERFs where no reference to, or comparison with, those earlier results, is given.

SPECIFIC COMMENTS (* = more major)

14: This conclusion is specific to the TLS

16: "becomes weaker at higher altitudes and at higher latitudes below 150 hPa." This is a bit ambiguous. Does this means heights at pressures below 150 hPa or heights below the height of the 150 hPa surface. These would have opposite meanings.

47-48: There is a slight overlap between this submitted paper and the paper published in ACP by the core PDRMIP team – Hodnebrog et al: https://doi.org/10.5194/acp-

19-12887-2019, which is not cited here. There is very little discussion of stratospheric water vapor in that paper, but nevertheless effects are implicit in some figures (including in the Supplemental) and so it should be referred to at appropriate points in the paper. In addition, a reader may wonder about the tropospheric wv response in PDRMIP models, and so it would be beneficial to point to that paper for that reason.

57: Presumably the 3xCH4 experiments have no resulting change in SWV due to the oxidation of additional methane?

*90 and many other places: There are repeated statements that there is no surface temperature response in the fixed SST runs, but this is not correct, with implications for the definition of ERF. See for example the fast surface temperature response in Smith et al. 2018, the alternative definitions of ERF in Richardson et al. 2019, and other PDRMIP papers.

129: I think it is necessary that a comparison of ERFs (and the associated feedback parameter) with Richardson et al. (including for the CFCs and N2O in their supplement) is presented both to confirm they are in reasonable agreement and also to make clear that the ERFs derived here are not original work with the PDRMIP output.

139: "tend to be larger" Isn't it clearly larger?

*148 and throughout: Rather little is said about intermodel differences. For example, on HADGEM3, more discussion of its apparent outlier status on some plots seems necessary. The text says it is "likely connected" to the larger surface warming, but it seems the climate sensitivity is about double the multi-model average but the slow SW response is around a factor of 4 larger. Is that because the TTL temperature change is 4 times higher (per unit ERF)? Another example is that apparently half the models have a slow SW response to BC of the opposite sign (Fig 1a) to the multimodel mean. Is there any obvious reason why? As far as I can see BC causes a warming in all models. One thing I miss from this study, and encourage the authors to look at if they have the resource, is the degree to which the model's background climatology of stratospheric

water vapor or TTL temperature could explain some of the intermodel differences.

156: Is this linear regression done once across all simulations and all perturbations. If not, I am unclear which perturbations have been used for the regression.

159: This is a relatively short paper and I wondered whether the supplementary figures could be brought into the main text?

171-172: This repeats a point already made at 141-142.

201: I am sorry if I miss it, but I see very little discussion of stratospheric temperature changes in the Jain et al paper. The role of CFCs on the vertical profile of temperature can be seen in many papers such as Forster et al. https://doi.org/10.1007/s003820050182 and Forster and Joshi 10.1007/s10584-005-5955-7 (by the way, I am not Forster!)

*204: This statement on shortwave radiation is strange. There may be a small shortwave effect from the reduced reflected flux from the troposphere, but there is a long history of simulations that clearly attribute the stratospheric cooling due to increased tropospheric ozone to the decreased upwelling thermal infrared radiation. E.g. Ramaswamy and Bowen https://doi.org/10.1029/94JD01310, Berntsen et al https://doi.org/10.1029/97JD02226 and the Forster et al. paper referred to above.

206: "Tropospheric O3 is also transported". As I understand it, ozone is imposed in the models and not advected. I don't know what this sentence means.

212: "larger than 50%". CAM5 and MPI-ESM look less than 50%?

*248: Returning to General Point#1, the Summary feels a very mechanical repetition of the results in the paper without any discussion of the wider implications, remaining uncertainties, or possible future avenues/priorities for improving understanding.

273: Strictly Fig 5 refers to TLS only

519-520: I think the markers are only reported when there are more than 3 contributing

models?

Typo

46L "responses" -> "responds"

Throughout: This may be common usage, but the paper refers thoughout to the ensemble mean when other papers would refer to its as the multi-model mean (ensemble could refer to different runs from the same model with perturbed initial conditions or physics). I dont have a strong opinion on this.
* * *

---

## Referee Comment (RC2) · Anonymous Referee #1 · 27 Jul 2020

This paper analyzes the changes of the stratospheric water vapor simulated by an ensemble of climate models under different forcing agents. The water vapor response is separated into fast and slow components, and tropical lower stratosphere and polar lowermost stratospheric regions are considered. It is found that the SST-mediated slow response generally dominates over the fast response, and shows a consistency across different forcing. The fast response, on the other hand is found to be linked to the fast adjustment of the cold point temperature to the climate forcing. This study provides a key answer to the question of how the stratospheric water vapor feedbacks to climate changes. I recommend publication of the paper after the authors address my generally minor comments.

[Figure]

1. Why the latitudinal bands of 30-50 are left out? It covers a considerably large area, and may be more subjective to the horizontal mixing than the polar region. Even it may be messy and don't show an as good consistency among forcing agents and across models as the polar regions or the tropical region, it still worth reporting. Furthermore, the 50S-90S may not be a good representation of the Southern Hemisphere extratropics. This is because many models suffer a too strong southern polar vortex and hence the simulated southern polar stratosphere is too isolated. This can be hinted from Fig. 4a and Fig. S4, where a clear barrier is seen near 60S.

2. The regression method to get the equilibrium water vapor response seems to be unnecessarily complicate, especially the results are not too different from the simple average of the last 30 years. The authors first fit the radiative flux and water vapor time series with an exponential function, then regress the the last 30 years of the fitted function. All these fitting and regression have potential introduce artificial biases and uncertainties. Recent studies also show that the ECS from the Gregory method may not be a good estimate of the true ECS (e.g. Winton et al. 2020). In addition, without a sufficiently long simulation, one can not validate whether the "equilibrium" from the regression is the true equilibrium. It makes more sense to me to simply use the average of the last 30 years while acknowledging that the models have not fully reached the equilibrium.

3. It may be worth pointing out how the PDRMIP model ensembles relate to the CMIP5 ensembles. From Fig. 2b, it seems that all of these models except HadGEM3 are on the weaker side of the CMIP5 ECS estimation range. I am also surprised to see that these models do not show an more distinct efficacy among different forcing agents (Hansen et al. 2005).

4. The authors relate the slow response to the surface temperature and relate the fast response to the cold point temperature. I believe the slow response would also be regulated by the cold point temperature. It may be interesting to show that if the relationship between the stratospheric water vapor and the cold point temperature holds

from the fast adjustment to the slow response.

5. While Fig. 3 shows a consistent relationship between stratospheric water vapor and global mean surface temperature across various forcing, the temperature sensitivity does not seem to be so consistent in Fig. S4. Much more stratospheric moistening is seen in response to the solar forcing than others given the same surface temperature warming. This discrepancy needs to be resolved.

Other even more minor comments:

Line 85-86: How does the averaged of fixed SST with baseline atmosphere compare to the average of the coupled baseline simulations.

Line 96: $y=c+ab^x$ -> $y=c+ab^{(-x)}$

Line 101: Fig. S1 was not showing what is stated here. It seems the intended Fig. S1 is missing.

Line 147: Fig. S2-4. -> Fig. S1-3

Line 167: Fig. S5 -> Fig. S4

Line 191: Does the long wave effect of the tropospheric ozone also contribute?

References:

Hansen, J. M. and coauthors, 2005: Efficacy of climate forcings. J. Geophys. Res., 110, D18104, doi:10.1029/2005JD005776.

Winton, M. and coaturhors, 2020: Climate sensitivity of GFDL's CM4.0. Journal of Advances in Modeling Earth Systems, 12, e2019MS001838. https://doi.org/10.1029/2019MS001838

---

## Author Comment (AC1) · 6 Aug 2020

Response to comment from Reviewer #2

General comments

1. *"The paper largely focuses on interpreting the multi-model responses. While this is of course useful, it stops short of relating the new understanding to any real-world changes in SWV... How much does this work help in understanding past and possible future SWV changes?"*

*"In particular, note there has been some discussion of how the PDRMIP BC perturbations compare to observations (Allan et al, https://doi.org/10.1038/s41612-019-0073-9)..."*

We will add a new figure and associated discussion to the paper. Figure R1 below (it will be assigned a different number in the revised manuscript) shows an analysis similar to that shown in Fig. 6 of Hodnebrog et al. (2019). We estimate historical $\Delta$SWV in the TLS between 1980-2010 by multiplying the multi-model multi-experiment mean sensitivity of the TLS $\Delta$SWV$_{slow}$ to surface temperature change by the historical surface temperature change to estimate the slow component of the historical $\Delta$SWV. We multiply the multi-model mean TLS $\Delta$SWV$_{fast}$/ERF by the historical radiative forcing for each selected PDRMIP experiment and sum it up to estimate the fast component of the historical $\Delta$SWV. The fast component of the historical $\Delta$SWV contributed by each historical forcing agent is shown in Fig. R1b below. The historical surface temperature change and radiative forcing data used in this analysis are listed in Table R1 below.

This calculation suggests that forcing since 1980 has increased TLS SWV by 0.51$\pm$0.16 ppmv (Fig. 1a). 64% of this is due to the fast response, mainly from black carbon. 36% is due to the slow response, although this is probably an overestimate because our sensitivity values are for long-term.

We have also evaluated the SWV sensitivity and SWV fast response over 35°N-45°N between 100-80 hPa to re-compute the 1980-2010 SWV change using the same method, which is 0.65$\pm$0.20 ppmv. This value shows reasonable agreement with the SWV increase measured by Hurst et al. (2011) of 0.71$\pm$0.26 ppmv over Boulder between 16-18 km over 1980-2010, although one must not read too much into the comparison to mid-latitude measurements.

Dessler et al. (2014) and Hegglin et al. (2014) argue that there is not a detectable trend over this period. Such a conclusion is not inconsistent with ours because any actual trend estimate has to contend with short-term interannual variability (i.e, like that from the QBO and Brewer-Dobson Circulation variability), which can mask a small trend. Our estimate of the trend is for long-term (100 year-run) and therefore internal variability has a small impact.

For the fast component of the estimated historical $\Delta$SWV, radiative forcing by BC plays the dominant role (Fig. R1b below). As pointed out by the reviewer, the radiative effect by BC in the PDRMIP is different from that shown in models using observationally constrained aerosol forcing, which may overestimate the heating in the UTLS region (Allen et al., 2019). However, Allen et al. (2019) also noted that uncertainties exist in their observationally constrained aerosol forcing. There are also uncertainties in the historical BC forcing listed in IPCC AR5 (Myhre et al. 2013). These bring in uncertainties in the fast component of the estimated historical $\Delta$SWV

by PDRMIP. These clearly merit more discussions in the future. We'll add this discussion about limitations and uncertainties of our PDRMIP estimates in the Summary and Discussion section.

**Table R1: Historical global average surface temperature change and radiative forcing (RF) by Greenhouse gases (GHGs) and halocarbons over 1980-2010.**

| $GMST^a$ (K) | 0.506 |
|---|---|
| Total $\Delta SWV$ (ppmv) | 0.51 |
| $\Delta SWV_{slow}$ (ppmv) | 0.18 |
| $\Delta SWV_{fast}$ (ppmv) | 0.32 |

| Forcing agents | RF ($Wm^{-2}$) | $\Delta SWV_{fast}$ by each forcing agent (ppmv) |
|---|---|---|
| $CO_2^b$ | 0.715 | 0.007 |
| $CH_4^c$ | 0.055 | 0.008 |
| $BC^d$ | 0.3 | 0.286 |
| CFC-12[e] | 0.068 | 0.015 |
| CFC-11[f] | 0.015 | 0.004 |
| $N_2O^g$ | 0.042 | 0.004 |

**a:** We used NOAA Merged Land Ocean Global Surface Temperature Analysis (NOAAGlobalTemp) V5 (Zhang et al. 2020) to compute the global surface temperature change. We use values averaged over 2005-2015 minus that averaged over 1975-1985.

**b, c, e, f, and g:** We compute the RFs using the formulae listed in Table 3 of Myhre et al. (1998). These formulae were also used to compute RFs of $CO_2$, $CH_4$, and $N_2O$ in IPCC reports (Myhre et al. 2013).

**b, c, and g:** Concentrations of GHGs were used to compute RFs. We use the $CO_2$, $CH_4$, and $N_2O$ samples collected in glass flasks at Cold Bay, Alaska, United States (CBA) from the ERSL GML website (Dlugokencky et al. 2020). For $CO_2$, concentrations averaged over 2005-2015 and averaged over 1978-1985 are used. For $CH_4$, concentrations averaged over 2005-2015 and averaged over 1983-1985 are used. For $N_2O$, concentration averaged over 2005-2015 was used, and 310 ppb (The IPCC 1990 and 1992 assessments) was used as a substitute for 1980 concentrations due to lack of data.

**e-f:** Concentrations of CFC-12 and CFC-11 were used to compute RFs. We use CFC-12 and CFC-11 data from combined stations from the NOAA/ESRL Global Monitoring Division. Concentrations averaged over 2005-2015 and averaged over 1977-1985 are used.

**d:** We use 0.4 $Wm^{-2}$, the BC RF between 1750-2011 reported in IPCC AR5, minus 0.1 $Wm^{-2}$, the BC RF between 1750-1993 reported in 1995 IPCC report (See Table 3 of Myhre et al. 2013). Note uncertainties for the IPCC AR5 BC RF is 0.05 to 0.8 $Wm^{-2}$, and uncertainties for the 1995 IPCC BC RF is 0.03 to 0.3$Wm^{-2}$.

[Figure]

Figure R1: (a) 1980-2010 TLS (30°S-30°N, 70 hPa) SWV change estimated using PDRMIP results. See texts above for method description. The uncertainty for the PDRMIP estimated SWV change is the sum of the uncertainty in the slow component and the uncertainty in the fast component. (b) The fast component of the PDRMIP-estimated 1980-2010 SWV change contributed by each historical forcing agent.

*2. "As noted in the specific comments, I feel that there is inadequate recognition that some of the results presented here are also presented, either explicitly or implicitly, in some earlier papers from the PDRMIP group – this is particularly so for the ERFs where no reference to, or comparison with, those earlier results, is given."*

Thanks for pointing this out. We'll add references to related results from earlier PDRMIP studies in the revised paper. Please see the responses to specific comments below related to previous PDRMIP studies:

*"47-48: There is a slight overlap between this submitted paper and the paper published in ACP by the core PDRMIP team – Hodnebrog et al: https://doi.org/10.5194/acp-19-12887-2019, which is not cited here..."*

We will reference results from Hodnebrog et al. (2019) in the revised paper.

*"129: I think it is necessary that a comparison of ERFs (and the associated feedback parameter) with Richardson et al. (including for the CFCs and N2O in their supplement) is presented both to confirm they are in reasonable agreement and also to make clear that the ERFs derived here are not original work with the PDRMIP output."*

We will add a statement making it clear that we are not the first to calculate ERFs with PDRMIP output. Table R2 (below) compares multi-model mean ERF with those listed in Richardson et al.

(2019); clearly, this comparison shows good agreement. In the revised manuscript, we will add this table comparing our values to Richardson's to the supplementary material.

The ERF in the submitted manuscript is computed using the same method as the "ERF$_{sst}$" in Richardson et al. (2019), so we directly compare multi-model mean values with their ERF$_{sst}$.

Table R2: Comparison of multi-model mean values with Richardson et al. (2019). In the parentheses, we list 95% confidence intervals obtained from Monte Carlo samples as described in Section 2.2 of the submitted manuscript for experiments that are performed by 6 or more models.

| | 2xCO$_2$ | 3xCH$_4$ | 2%Solar | 10xBC | 5xSO$_4$ | 10xCFC-12 | 10xCFC-11 | 3xN$_2$O | 5xO$_3$ | 10xBCSLT |
|---|---|---|---|---|---|---|---|---|---|---|
| ERF$_{sst}$ (Wm$^{-2}$) from Richardson et al. (2019) | 3.71±0.30 | 1.15±0.25 | 4.17±0.13 | 1.18 ±0.75 | - 3.71±1.94 | 1.39 (1.21 to 1.54) | 1.19 (1.17 to 1.21) | 1.60 (1.23 to 2.14) | 3.47 (2.45 to 4.49) | 1.10 |
| ERF (Wm$^{-2}$) | 3.75 (3.58 to 3.92) | 1.20 (1.02 to 1.38) | 4.23 (4.15 to 4.33) | 1.21 (0.68 to 1.93) | -3.68 (-5.28 to -2.63) | 1.35 | 1.17 | 1.80 | 3.45 | 1.21 |

Specific comments

*"14: This conclusion is specific to the TLS"*

Yes, we agree with this, although the cold point temperature does have *some* influence in the lowermost SWV(Dessler et al., 1995). But the control is not as strong as that in the TLS (see Fig. 6b-c in submitted manuscript) and the lowermost SWV is controlled by multiple factors. We'll edit the text in the revised manuscript to make sure there is no confusion.

*"16: "becomes weaker at higher altitudes and at higher latitudes below 150 hPa." This is a bit ambiguous. Does this means heights at pressures below 150 hPa or heights below the height of the 150 hPa surface. These would have opposite meanings."*

It means altitudes below the 150 hPa surface. We'll edit the text in the revised manuscript to avoid confusion.

*"57: Presumably the 3xCH4 experiments have no resulting change in SWV due to the oxidation of additional methane?"*

Yes, indirect chemical effects are not included in the 3xCH$_4$ experiment. We will add a sentence saying this to the revised manuscript.

*"90 and many other places: There are repeated statements that there is no surface temperature response in the fixed SST runs, but this is not correct, with implications for the definition of ERF."*

Yes, the reviewer is correct that land surface temperatures can respond to the forcing. We'll edit the text in the revised manuscript and make sure there is no confusion.

*"139: tend to be larger" Isn't it clearly larger?*

Yes, the reviewer is correct. We'll edit this text in the revised manuscript.

*"*148 and throughout: Rather little is said about intermodel differences. For example, on HADGEM3, more discussion of its apparent outlier status on some plots seems necessary. The text says it is "likely connected" to the larger surface warming, but it seems the climate sensitivity is about double the multi-model average but the slow SW response is around a factor of 4 larger. Is that because the TTL temperature change is 4 times higher (per unit ERF)?"*

To answer the question about the HadGEM3 model, Figure R2 below shows the equilibrium slow response of TTL temperature ($\Delta T_{slow}$) per unit ERF. The HadGEM3 $\Delta T_{slow}$/ERF is between 2.64-3.97 times the multi-model mean $\Delta T_{slow}$/ERF for experiments $2xCO_2$, $3xCH_4$, 2%Solar, 10xBC, and 10xCFC-12. Figure R3 below shows the vertical profile of tropical atmospheric temperature slow response ($\Delta T_{slow}$) for $2xCO_2$. All models show a maximum warming in the upper troposphere. Since the surface warming in HadGEM3 is larger than all other models, its upper tropospheric warming is also largest. Longwave radiation emitted from the upper troposphere warms the TTL level (Lin et al., 2017), so the larger upper tropospheric warming in HadGEM3 also results in larger TTL heating than other models. The relationship between surface warming and TTL warming is not linear.

That said, we cannot conclusively identify a cause given the information archived. So we will remove the claim that the difference is likely connected to surface warming and add a sentence saying more work on the causes of these differences is warranted.

*"Another example is that apparently half the models have a slow SW response to BC of the opposite sign (Fig 1a) to the multi-model mean. Is there any obvious reason why? As far as I can see BC causes a warming in all models."*

Figure R4 below shows the vertical profile of tropical temperature slow (a) and fast (b) responses per unit ERF for the 10xBC experiment. The 10xBC does cause a warming at the surface and in the troposphere due to a positive TOA ERF in all models. In the TTL and lower stratosphere (LS), however, the heating is mainly caused by the fast adjustment (Fig. R4b below). The slow

temperature response in the TTL is the residual of the total response minus the fast adjustment, which is negative or close to zero (Fig. R4a below).

It is therefore our contention that some of these negative values are artifacts of the method we use to estimate equilibrium response. Support for this comes from Fig. 3 of the paper. The values in this figure come from regressions of $\Delta SWV_{slow}$ vs. $\Delta Ts$ in the BC runs. This method does not require differencing two large numbers, so we feel it is more robust. It shows that most models have a positive response of SWV due to BC-induced warming. For those models that produce negative slopes for $\Delta SWV_{slow}$ vs. $\Delta Ts$ in the BC runs, there is large uncertainty in the regression, because the surface temperature change in those models are small. We will note this in the revised manuscript.

[Figure]

Figure R2: Equilibrium slow response of TTL temperature (100 hPa, averaged between 30°N-30°S) per ERF for all models and perturbations.

[Figure]

Figure R3: Vertical profile of atmospheric temperature equilibrium slow response normalized by ERF (K·(Wm$^{-2}$)$^{-1}$) for 2xCO$_2$. Temperature is averaged over 30°N-30°S.

[Figure]

Figure R4: Profiles of equilibrium slow (a) and fast (b) temperature response for the 10xBC experiment, normalized by ERF (K·(Wm$^{-2}$)$^{-1}$), and averaged over 30°N-30°S. The color coding indicates results from different models.

*"One thing I miss from this study, and encourage the authors to look at if they have the resource, is the degree to which the model's background climatology of stratospheric water vapor or TTL temperature could explain some of the intermodel differences."*

We have investigated the SWV in the fixed SST baseline simulations. Based on our analyses, the baseline climatology SWV does not explain the inter-model differences in the responses to forcing agents. As an example, Fig. R5 below shows the TLS SWV slow response (first row) and TTL temperature slow response (second row) vs. the baseline TLS SWV climatology and baseline TTL temperature climatology. We omitted $3xN_2O$, $5xO_3$, and 10xBCSLT, because fewer than three models performed these experiments. There is no correlation between the SWV and temperature slow responses and the baseline climatology. In particular, HadGEM3 produces extremely large slow responses for most experiments, however, in Fig. R5 below, its baseline SWV and temperature climatology is not the largest among the models.

[Figure]

Figure R5: Top row: The TLS SWV slow response (ppmv) vs. the baseline TLS SWV climatology (ppmv). Bottom row: The TTL temperature slow response (K) (100 hPa, averaged over 30°N-30°S) vs. the baseline TTL temperature climatology (K) (100 hPa, averaged over 30°N-30°S). The baseline climatology is obtained from the fixed SST simulations averaged over the last 10 years.

*"156: Is this linear regression done once across all simulations and all perturbations. If not, I am unclear which perturbations have been used for the regression."*

We regressed the annual mean $\Delta SWV_{slow}$ time series vs the annual mean $\Delta Ts$ for each perturbation and model separately. We showed the scatter plot for each perturbation and model in Figs. S1-3 of the submitted manuscript and showed the slopes of the regression done for each perturbation and model in Fig. 3 of the submitted manuscript.

*"159: This is a relatively short paper and I wondered whether the supplementary figures could be brought into the main text?"*

It remains our opinion that the key figures are included in the paper. Thus, in order to keep the take-home message concise, we have left the content of the supplement unchanged.

*"171-172: This repeats a point already made at 141-142."*

We'll edit the text in the revised manuscript to make sure there is no such repetition.

*"201: I am sorry if I miss it, but I see very little discussion of stratospheric temperature changes in the Jain et al paper. The role of CFCs on the vertical profile of temperature can be seen in many papers such as Forster et al. https://doi.org/10.1007/s003820050182 and Forster and Joshi 10.1007/s10584-005- 5955-7…"*

In the submitted paper where we discussed the radiative heating in the UTLS by CFCs, we were referring to the text in Section 3.3 of Jain et al. (2013), where they stated that "Halocarbons absorb predominantly in the window region (750-1250 $cm^{-1}$), in the linear line limit; therefore in the stratosphere they absorb the upwelling radiation from the troposphere and increase the heating rate of the stratosphere". Forster and Joshi (2005) also pointed out that, compared to $CO_2$, halocarbons preferentially warm the TTL, rather than the troposphere.

We agree with the reviewer that it is useful to reference papers that explicitly investigated vertical temperature profiles forced by CFCs. We'll add these references in the revised manuscript.

*"*204: This statement on shortwave radiation is strange. There may be a small shortwave effect from the reduced reflected flux from the troposphere, but there is a long history of simulations that clearly attribute the stratospheric cooling due to increased tropospheric ozone to the decreased upwelling thermal infrared radiation. E.g. Ramaswamy and Bowen https://doi.org/10.1029/94JD01310, Berntsen et al https://doi.org/10.1029/97JD02226 and the Forster et al. paper referred to above."*

Thanks for pointing this out. We'll edit the text to say, "Increases of tropospheric $O_3$ ($5xO_3$) reduce the upwelling longwave radiation, which cools the stratosphere. The longwave radiation absorbed heat the TTL region…".  We will also add references to these papers.

*"206: "Tropospheric O3 is also transported". As I understand it, ozone is imposed in the models and not advected. I don't know what this sentence means."*

This is correct: In the 5xO$_3$ experiment, the PDRMIP group used 5 times the tropospheric ozone distribution (TROP) in the paper by MacIntosh et al. (2016). We will correct this statement in the paper.

*"212: "larger than 50%". CAM5 and MPI-ESM look less than 50%?"*

Yes, this was poorly worded. We have completely re-written the paragraph, so it now read: The 3xCH$_4$ also includes some models that produce large TLS $\Delta SWV_{fast}$/ERF magnitudes. This is likely due to TTL heating by 3xCH$_4$ (Figure 5) due to the shortwave absorption by CH$_4$, which is explicitly treated in some models, including CAM5, CanESM2, MPI-ESM, and MIROC-SPRINTARS (Smith et al., 2018). These models are also the ones that produce the largest TLS $\Delta SWV_{fast}$ contributions.

*"*248: Returning to General Point#1, the Summary feels a very mechanical repetition of the results in the paper without any discussion of the wider implications, remaining uncertainties, or possible future avenues/priorities for improving understanding."*

We will add Fig. R1 and related discussions above in the Discussion and Summary section. We will discuss that estimated historical SWV change shows reasonable agreement with existing observed SWV record. We'll also discuss its uncertainties, which include the uncertainties in the radiative effect of BC forcing and the lack of long-term observation record of SWV as a reference.

*"273: Strictly Fig 5 refers to TLS only"*

We'll edit the text and make sure there will be no such confusion in the revised manuscript.

*"519-520: I think the markers are only reported when there are more than 3 contributing models?"*

Yes, the multi-model mean and error bars are shown for perturbations that are performed by more than three models. We will mention this in the revised caption.

*"46L "responses" -> "responds""*

We'll modify the text in the revised manuscript.

*"Throughout: This may be common usage, but the paper refers throughout to the ensemble mean when other papers would refer to it as the multi-model mean (ensemble could refer to different runs from the same model with perturbed initial conditions or physics…"*

Thanks for pointing this out. To avoid confusion, we'll use multi-mode mean in the revised manuscript.

**References:**

[revised manuscript text omitted]

---

## Short Comment (SC1) · 17 Aug 2020

Response to comment from Reviewer #1

*"1. Why the latitudinal bands of 30-50 are left out? It covers a considerably large area, and may be more subjective to the horizontal mixing than the polar region. Even it may be messy and don't show an as good consistency among forcing agents and across models as the polar regions or the tropical region, it still worth reporting. Furthermore, the 50S-90S may not be a good representation of the Southern Hemisphere extratropics. This is because many models suffer a too strong southern polar vortex and hence the simulated southern polar stratosphere is too isolated. This can be hinted from Fig. 4a and Fig. S4, where a clear barrier is seen near 60S."*

We first note that the response of $\Delta SWV_{slow}$ to surface temperature as a function of latitude is plotted in Fig. 5a of the submitted manuscript. We also reported the regression slope of $\Delta SWV_{fast}$ vs cold point temperature fast response in Fig. 5b of the submitted manuscript.

However, to more clearly answer the reviewer's question, Figure R1 below shows the equilibrium $\Delta SWV_{slow}$ and $\Delta SWV_{fast}$ and their contribution to the total equilibrium $\Delta SWV$ for water vapor averaged at 200 hPa 30°N-50°N and 30°S-50°S. In Figure R2 below, we also show the slope of $\Delta SWV_{slow}$ annual mean time series vs surface temperature time series for water vapor averaged at 200 hPa 30°N-50°N and 30°S-50°S. We will include both of these figures in the supplement. The results in the 30-50 degree latitude band show larger magnitudes compared to the results in the 50-90 degree latitude band. However, our major conclusions remain the same: The slow response plays a dominant role and contributes to close to 100% of the total response for most perturbations; The sensitivity shows general agreement across different perturbations.

[Figure]

Figure R1: Same as Figure 1 in the submitted paper, but for 200 hPa 30°N-50°N (a-d) and 200 hPa 30°S-50°S (e-h).

[Figure]

Figure R2: Same as Figure 3, but for 200 hPa 30°N-50°N (a) and 200 hPa 30°S-50°S (b).

*"2. The regression method to get the equilibrium water vapor response seems to be unnecessarily complicate, especially the results are not too different from the simple average of the last 30 years. The authors first fit the radiative flux and water vapor time series with an exponential function, then regress the last 30 years of the fitted function. All these fitting and regression have potential introduce artificial biases and uncertainties. Recent studies also show that the ECS from the Gregory method may not be a good estimate of the true ECS (e.g. Winton et al. 2020). In addition, without a sufficiently long simulation, one can not validate whether the "equilibrium" from the regression is the true equilibrium. It makes more sense to me to simply use the average of the last 30 years while acknowledging that the models have not fully reached the equilibrium."*

We initially did approximate equilibrium ΔSWV using averages of the last 30 years of the runs. However, we analyzed one model that was run for 2600 years and found that the last 30 years of a 100- or 150-year run significantly underestimated the equilibrium. Thus, we developed the method that we presently use in the paper to better produce equilibrium estimates and validated it in the 2600-year model run, which is close to its equilibrium climate state.

We will include more details of this validation in the revised manuscript. Our method gave results in reasonable agreement with the climate model at equilibrium: "We analyzed runs of the fully coupled Max Planck Institute Earth System Model version 1.1 (MPI-ESM1.1) (Maher et al., 2019), which has a transient climate response and an effective climate sensitivity near the

middle of the CMIP5 ensemble range (Adams and Dessler, 2019; Dessler, 2020). It includes a 2000-year preindustrial control run and a 2614-year abruptly quadrupled $CO_2$ run. The values of $\Delta$SWV averaged over the last 30 years of the $4xCO_2$ run relative to the control run are 4.60 ppmv in the TLS, 22.40 ppmv in the NH LMS, and 9.69 ppmv in the SH LMS. We expect this to be close to equilibrium $\Delta$SWV because the trend in global average surface temperature over the last 500 years of the $4xCO_2$ run is 0.02 K per century. We use the regression method to estimate the equilibrium $\Delta$SWV using MPI-ESM1.1 water vapor mixing ratio time series over the first 100 years and obtain estimates of 4.38 ppmv in the TLS, 20.01 ppmv in the NH LMS, and 9.07 ppmv in the SH LMS; these yield differences of 0.22 ppmv in the TLS, 2.39 ppmv in the NH LMS, and 0.62 ppmv in the SH LMS. Thus, our method underestimates the true equilibrium value by 5% in the TLS, 11% in the NH LMS, and 6% in the SH LMS."

We will also produce estimates using the last 30 years, and we will add those to a table in the supplementary material.

*"3. It may be worth pointing out how the PDRMIP model ensembles relate to the CMIP5 ensembles. From Fig. 2b, it seems that all of these models except HadGEM3 are on the weaker side of the CMIP5 ECS estimation range. I am also surprised to see that these models do not show an more distinct efficacy among different forcing agents (Hansen et al. 2005)."*

The PDRMIP models are a subset of the CMIP5 models. The PDRMIP multi-model average ECS we estimated is 3.6 K, which is 10% larger than the whole CMIP5 ensemble (3.3 K) (Zelinka et al. 2020). We will add a statement to the revised manuscript comparing the PDRMIP models' ECS to that in the CMIP5 ensemble.

As far as forcing efficacy goes, Hansenet al. (2005) also pointed out that efficacies depend on the method of which radiative forcing is defined. A more recent paper by Richardson et al. (2019) (which we already referenced in the submitted manuscript) using PDRMIP data showed that forcing efficacies calculated from effective radiative forcing have values close to one. Our results are in good agreement with that paper.

*"4. The authors relate the slow response to the surface temperature and relate the fast response to the cold point temperature. I believe the slow response would also be regulated by the cold point temperature. It may be interesting to show that if the relationship between the stratospheric water vapor and the cold point temperature holds from the fast adjustment to the slow response."*

It certainly may be the case that the slow response is mediated by TTL temperatures, but by no means is that certain. Dessler et al. (2016) showed that, in two climate models, at least, a significant fraction of the long-term trend (and slow response) was due to increases in convective moistening, which bypasses the TTL cool trap.

We have done analyses testing whether the PDRMIP models and experiments show agreement for the relation between $\Delta SWV_{slow}$ vs the CPT slow response (Figure R3). Results from the models and experiments show good agreement. The slope is 0.72 ppmv/K, which is larger than the slope obtained from the fast response. Nevertheless, correlation does not prove causality and this result could arise from either TTL control or if convective moistening also correlates with the CPT slow response, or some combination. We will add a sentence to the paper describing this analysis.

[Figure]

Figure R3: Same as Figure 6a of the submitted manuscript, but for TLS SWV slow response vs the CPT slow response.

*"5. While Fig. 3 shows a consistent relationship between stratospheric water vapor and global mean surface temperature across various forcing, the temperature sensitivity does not seem to be so consistent in Fig. S4. Much more stratospheric moistening is seen in response to the solar forcing than others given the same surface temperature warming. This discrepancy needs to be resolved."*

We list the regression slope values in Table R4 below. These values are the same as we have shown in Fig. 3 of the submitted manuscript. It may not be clear in Fig. 3 of the submitted manuscript, but it is clear in Table R4 below that the sensitivities are indeed larger in some experiments, such as the 2%Solar experiment. This is the same for slopes computed using the two different units – ppmv/K and %/K (listed in the parentheses).

We will add Table R4 to the supplement of the revised manuscript.

Table R4: Regression slope of $\Delta SWV_{slow}$ annual mean time series vs surface temperature change annual mean time series for all perturbations and models. The unit of the values is ppmv/K. The unit of the values in parentheses is %/K.

**TLS SWV**

| | 2xCO2 | 3xCH4 | 2%SOLAR | 10xBC | 5xSO4 | 10xCFC12 | 10xCFC11 | 3xN2O | 5xO3 | 10xBCTAU |
|---|---|---|---|---|---|---|---|---|---|---|
| CAM4 | 0.26 (6.35) | 0.25 (6.19) | 0.30 (7.27) | 0.34 (8.44) | 0.31 (7.64) | 0.27 (6.50) | nan | 0.29 (7.04) | nan | 0.18 (4.44) |
| CAM5 | 0.36 (9.74) | 0.26 (6.98) | 0.36 (9.76) | 0.11 (3.05) | 0.11 (3.06) | 0.23 (6.23) | nan | nan | nan | nan |
| HadGEM3 | 0.92 (17.89) | 0.64 (12.47) | 1.01 (19.64) | 0.54 (10.48) | 0.40 (7.86) | 0.70 (13.56) | nan | nan | nan | nan |
| HadGEM2 | 0.46 (9.22) | 0.43 (8.52) | 0.60 (11.94) | 1.78 (35.29) | 0.35 (6.95) | 0.63 (12.49) | 0.67 (13.34) | 0.66 (13.05) | nan | nan |
| CanESM2 | 0.27 (10.25) | -0.01 (-0.23) | 0.38 (14.59) | 0.12 (4.68) | 0.12 (4.77) | nan | nan | nan | nan | nan |
| IPSL-CM5A | 0.62 (26.79) | 0.36 (15.52) | 0.79 (33.70) | 0.52 (22.30) | 0.28 (11.78) | nan | nan | nan | nan | nan |
| MPI-ESM | 0.53 (12.78) | 0.22 (5.22) | 0.54 (12.95) | nan | nan | nan | nan | nan | nan | nan |
| GISS | 0.28 (16.28) | 0.16 (9.05) | 0.31 (18.17) | 0.25 (14.80) | 0.20 (11.73) | 0.18 (10.23) | nan | nan | nan | 0.15 (8.53) |
| MIROC-SPRINTARS | 0.08 (2.61) | 0.11 (4.07) | 0.14 (5.04) | 0.09 (3.15) | 0.09 (3.26) | 0.14 (5.06) | 0.10 (3.64) | 0.08 (2.83) | 0.05 (1.59) | nan |
| ensemble average | 0.42 (12.43) | 0.27 (7.53) | 0.49 (14.78) | 0.47 (12.77) | 0.23 (7.13) | 0.36 (9.01) | 0.39 (8.49) | 0.34 (7.64) | 0.05 (1.59) | 0.16 (6.49) |

**NH LMS SWV**

| | 2xCO2 | 3xCH4 | 2%SOLAR | 10xBC | 5xSO4 | 10xCFC12 | 10xCFC11 | 3xN2O | 5xO3 | 10xBCTAU |
|---|---|---|---|---|---|---|---|---|---|---|
| CAM4 | 2.67 (11.63) | 1.55 (6.71) | 2.47 (10.75) | 2.41 (10.49) | 1.91 (8.50) | 2.08 (8.84) | nan | 1.22 (5.18) | nan | 0.58 (2.44) |
| CAM5 | 1.67 (7.70) | -0.26 (-0.59) | 1.77 (8.28) | -0.35 (-0.75) | 0.90 (4.35) | 0.30 (1.85) | nan | nan | nan | nan |
| HadGEM3 | 4.31 (27.80) | 3.19 (20.72) | 5.15 (33.01) | 3.84 (24.77) | 1.60 (10.22) | 3.49 (22.56) | nan | nan | nan | nan |
| HadGEM2 | 2.17 (20.79) | 1.71 (16.08) | 2.44 (23.34) | 3.99 (38.14) | 1.20 (11.47) | 2.08 (19.87) | 2.03 (19.42) | 2.02 (19.33) | nan | nan |
| CanESM2 | 3.18 (22.65) | 2.61 (18.76) | 3.37 (23.85) | 3.59 (25.94) | 2.00 (14.31) | nan | nan | nan | nan | nan |
| IPSL-CM5A | 2.42 (16.09) | 1.94 (13.21) | 2.81 (18.59) | 2.38 (15.88) | 1.94 (13.05) | nan | nan | nan | nan | nan |
| MPI-ESM | 2.56 (12.29) | 1.56 (7.45) | 2.36 (11.26) | nan | nan | nan | nan | nan | nan | nan |
| GISS | 1.19 (15.55) | 0.72 (8.90) | 1.12 (14.45) | 1.05 (13.67) | 0.73 (8.84) | 0.76 (9.06) | nan | nan | nan | 0.93 (11.54) |
| MIROC-SPRINTARS | 2.45 (19.18) | 2.30 (19.20) | 2.40 (18.69) | 2.81 (22.27) | 1.85 (14.13) | 3.00 (23.59) | 2.42 (19.15) | 2.20 (17.99) | 3.36 (24.67) | nan |
| ensemble average | 2.52 (17.08) | 1.70 (12.27) | 2.65 (18.02) | 2.46 (18.80) | 1.52 (10.61) | 1.95 (14.29) | 2.23 (19.29) | 1.82 (14.17) | 3.36 (24.67) | 0.76 (6.99) |

**SH LMS SWV**

| SH LMS SWV | 2xCO2 | 3xCH4 | 2%SOLAR | 10xBC | 5xSO4 | 10xCFC12 | 10xCFC11 | 3xN2O | 5xO3 | 10xBCTAU |
|---|---|---|---|---|---|---|---|---|---|---|
| CAM4 | 1.67 (11.05) | 1.53 (10.87) | 1.88 (12.55) | 0.61 (4.70) | 0.84 (5.44) | 1.76 (11.75) | nan | 1.21 (8.44) | nan | 0.84 (6.39) |
| CAM5 | 0.87 (5.60) | -0.14 (0.47) | 0.99 (6.22) | -0.53 (-2.24) | 0.34 (2.68) | -0.38 (-1.03) | nan | nan | nan | nan |
| HadGEM3 | 2.65 (22.08) | 1.63 (14.66) | 3.05 (25.21) | 0.90 (8.17) | 1.15 (9.91) | 1.75 (14.81) | nan | nan | nan | nan |
| HadGEM2 | 0.93 (12.88) | 0.61 (8.59) | 0.94 (13.33) | 1.08 (15.47) | 0.58 (8.10) | 0.63 (9.15) | 0.60 (8.67) | 0.78 (10.96) | nan | nan |
| CanESM2 | 1.24 (12.11) | 0.64 (6.68) | 1.36 (13.02) | 1.50 (14.70) | 0.85 (8.19) | nan | nan | nan | nan | nan |
| IPSL-CM5A | 1.60 (15.84) | 1.08 (11.08) | 1.74 (16.82) | 0.63 (6.53) | 1.11 (11.64) | nan | nan | nan | nan | nan |
| MPI-ESM | 1.34 (9.87) | 0.80 (6.12) | 1.45 (10.76) | nan | nan | nan | nan | nan | nan | nan |
| GISS | 0.65 (11.44) | -0.09 (-0.27) | 0.80 (14.48) | 0.00 (1.15) | 0.21 (4.54) | 0.01 (1.61) | nan | nan | nan | 0.32 (5.85) |
| MIROC-SPRINTARS | 1.16 (10.88) | 0.82 (7.72) | 1.14 (10.63) | 0.80 (8.10) | 0.88 (7.93) | 1.29 (11.97) | 0.78 (7.49) | 0.80 (7.81) | 1.34 (12.30) | nan |
| ensemble average | 1.35 (12.42) | 0.76 (7.33) | 1.48 (13.67) | 0.62 (7.07) | 0.75 (7.31) | 0.84 (8.04) | 0.69 (8.08) | 0.93 (9.07) | 1.34 (12.30) | 0.58 (6.12) |

*"Line 85-86: How does the averaged of fixed SST with baseline atmosphere compare to the average of the coupled baseline simulations."*

For TLS SWV, the difference between fixed SST baseline simulation and coupled baseline simulation is on the order of $0.01 - 1$ ppmv. For LMS SWV, the difference between fixed SST baseline simulation and coupled baseline simulation is on the order of $0.1 - 1$ ppmv.

The results are averaged over the entire period of the baseline simulations for both fixed SST run and coupled run.

*"Line 96: y=c+ab^x -> y=c+ab^(-x)*

*Line 101: Fig. S1 was not showing what is stated here. It seems the intended Fig. S1 is missing.*

*Line 147: Fig. S2-4. -> Fig. S1-3*

*Line 167: Fig. S5 -> Fig. S4"*

We have updated these in the manuscript.

*"Line 191: Does the long wave effect of the tropospheric ozone also contribute?"*

Yes, the tropospheric ozone has the long wave radiative effect. We are going to edit the text to "Increases of tropospheric $O_3$ ($5xO_3$) reduce the upwelling longwave radiation through longwave absorption, which cools the stratosphere…"

**References**

Adams, B. K. and Dessler, A. E.: Estimating Transient Climate Response in a Large-Ensemble Global Climate Model Simulation, Geophys. Res. Lett., 46(1), 311–317, doi:10.1029/2018GL080714, 2019.

Banerjee, A., Chiodo, G., Previdi, M., Ponater, M., Conley, A. J. and Polvani, L. M.: Stratospheric water vapor: an important climate feedback, Clim. Dyn., doi:10.1007/s00382-019-04721-4, 2019.

Dessler, A. E.: Potential Problems Measuring Climate Sensitivity from the Historical Record, J. Clim., 33(6), 2237–2248, doi:10.1175/JCLI-D-19-0476.1, 2020.

Dessler, A. E., Schoeberl, M. R., Wang, T., Davis, S. M. and Rosenlof, K. H.: Stratospheric water vapor feedback, Proc. Natl. Acad. Sci., 110(45), 18087–18091, doi:10.1073/pnas.1310344110, 2013.

Dessler, A. E., Ye, H., Wang, T., Schoeberl, M. R., Oman, L. D., Douglass, A. R., Butler, A. H., Rosenlof, K. H., Davis, S. M. and Portmann, R. W.: Transport of ice into the stratosphere and the humidification of the stratosphere over the 21st century, Geophys. Res. Lett., 43(5), 2323–2329, doi:10.1002/2016GL067991, 2016.

Fueglistaler, S.: Stratospheric water vapor predicted from the Lagrangian temperature history of air entering the stratosphere in the tropics, J. Geophys. Res., 110(D8), D08107, doi:10.1029/2004JD005516, 2005.

Fueglistaler, S., Dessler, A. E., Dunkerton, T. J., Folkins, I., Fu, Q. and Mote, P. W.: Tropical tropopause layer, Rev. Geophys., 47, 1–31, doi:10.1029/2008RG000267, 2009.

Hansen, J.: Efficacy of climate forcings, J. Geophys. Res., 110(D18), D18104, doi:10.1029/2005JD005776, 2005.

Maher, N., Milinski, S., Suarez-Gutierrez, L., Botzet, M., Dobrynin, M., Kornblueh, L., Kröger, J., Takano, Y., Ghosh, R., Hedemann, C., Li, C., Li, H., Manzini, E., Notz, D., Putrasahan, D., Boysen, L., Claussen, M., Ilyina, T., Olonscheck, D., Raddatz, T., Stevens, B. and Marotzke, J.: The Max Planck Institute Grand Ensemble: Enabling the Exploration of Climate System Variability, J. Adv. Model. Earth Syst., 11(7), 2050–2069, doi:10.1029/2019MS001639, 2019.

Mote, P. W., Rosenlof, K. H., McIntyre, M. E., Carr, E. S., Gille, J. C., Holton, J. R., Kinnersley, J. S., Pumphrey, H. C., Russell III, J. M. and Waters, J. W.: An atmospheric tape recorder: The imprint of tropical tropopause temperatures on stratospheric water vapor, J. Geophys. Res., 101(D2), 3989–4006, doi:10.1029/95JD03422, 1996.

Richardson, T. B., Forster, P. M., Smith, C. J., Maycock, A. C., Wood, T., Andrews, T., Boucher, O., Faluvegi, G., Fläschner, D., Hodnebrog, Ø., Kasoar, M., Kirkevåg, A., Lamarque, J. -F., Mülmenstädt, J., Myhre, G., Olivié, D., Portmann, R. W., Samset, B. H., Shawki, D., Shindell, D., Stier, P., Takemura, T., Voulgarakis, A. and Watson-Parris, D.: Efficacy of Climate Forcings in PDRMIP Models, J. Geophys. Res. Atmos., 124(23), 12824–12844, doi:10.1029/2019JD030581, 2019.

Zelinka, M. D., T. A. Myers, D. T. McCoy, S. Po-Chedley, P. M. Caldwell, P. Ceppi, S. A. Klein, and K. E. Taylor, 2020: Causes of higher climate sensitivity in CMIP6 models, Geophys. Res. Lett., 47, doi:10.1029/2019GL085782.

---

## Author Response (AR2)

**Response to reviewers**

We thank both reviewers for their useful comments on our paper. Please note that all line numbers refer to the version with *tracked changes*. Many of the comments below reproduce our previously uploaded responses to the reviewers' comments. This document, however, provides more detailed information about how we have modified our manuscript.

**Reviewer #1**

*"1. Why the latitudinal bands of 30-50 are left out? It covers a considerably large area, and may be more subjective to the horizontal mixing than the polar region. Even it may be messy and don't show an as good consistency among forcing agents and across models as the polar regions or the tropical region, it still worth reporting. Furthermore, the 50S-90S may not be a good representation of the Southern Hemisphere extratropics. This is because many models suffer a too strong southern polar vortex and hence the simulated southern polar stratosphere is too isolated. This can be hinted from Fig. 4a and Fig. S4, where a clear barrier is seen near 60S."*

We first note that the response of $\Delta SWV_{slow}$ to surface temperature as a function of latitude is plotted in Fig. 4a of the submitted manuscript. We also reported the regression slope of $\Delta SWV_{fast}$ vs cold point temperature fast response in Fig. 4b of the submitted manuscript.

However, to more clearly answer the reviewer's question, we have included results at 200 hPa between 30°N and 50°N in the revised supplement. Figure S1 in the revised supplement shows the equilibrium $\Delta SWV_{slow}$ and $\Delta SWV_{fast}$ and their contribution to the total equilibrium $\Delta SWV$ for water vapor averaged at 200 hPa 30°N-50°N and 30°S-50°S. Figure S2 in the revised supplement shows the slope of $\Delta SWV_{slow}$ annual mean time series vs surface temperature time series for water vapor averaged at 200 hPa 30°N-50°N and 30°S-50°S.

In the revised manuscript, we mentioned these results in lines 180-184 and lines 222-223. Our major conclusions remain the same: The slow response plays a dominant role and contributes to close to 100% of the total response for most perturbations; The sensitivity shows general agreement across different perturbations.

*"2. The regression method to get the equilibrium water vapor response seems to be unnecessarily complicate, especially the results are not too different from the simple average of the last 30 years. The authors first fit the radiative flux and water vapor time series with an exponential function, then regress the last 30 years of the fitted function. All these fitting and regression have potential introduce artificial biases and uncertainties. Recent studies also show that the ECS from the Gregory method may not be a good estimate of the true ECS (e.g. Winton et al. 2020). In addition, without a sufficiently long simulation, one can not validate whether the "equilibrium" from the regression is the true equilibrium. It makes more sense to me to simply use the average of the last 30 years while acknowledging that the models have not fully reached the equilibrium."*

In an early draft of this manuscript, we approximated equilibrium ΔSWV using averages of the last 30 years of the runs. However, we analyzed one model that was run for 2600 years and found that the last 30 years of a 100- or 150-year run significantly underestimated the equilibrium. Thus, we developed the method that we presently use in the paper to better produce equilibrium estimates and validated it in the 2600-year model run, which is close to its equilibrium climate state. Details of this validation are described in lines 130-140 in the revised manuscript.

However, in response to this comment, we have listed slow response estimated by averaging over the last 30 years of the coupled simulation in Table S2 of the revised supplement.

*"3. It may be worth pointing out how the PDRMIP model ensembles relate to the CMIP5 ensembles. From Fig. 2b, it seems that all of these models except HadGEM3 are on the weaker side of the CMIP5 ECS estimation range. I am also surprised to see that these models do not show an more distinct efficacy among different forcing agents (Hansen et al. 2005)."*

We have added a statement to the revised manuscript comparing the PDRMIP models' ECS to that in the CMIP5 ensemble in lines 64-65.

As far as forcing efficacy goes, Hansenet al. (2005) also pointed out that efficacies depend on the method of which radiative forcing is defined. A more recent paper by Richardson et al. (2019) (which we already referenced in the submitted manuscript) using PDRMIP data showed that forcing efficacies calculated from effective radiative forcing have values close to one. Our results are in good agreement with Richardson et al. (2019) (Table S3 in the revised supplement).

*"4. The authors relate the slow response to the surface temperature and relate the fast response to the cold point temperature. I believe the slow response would also be regulated by the cold point temperature. It may be interesting to show that if the relationship between the stratospheric water vapor and the cold point temperature holds from the fast adjustment to the slow response."*

It certainly may be the case that the slow response is mediated by TTL temperatures, but by no means is that certain. Dessler et al. (2016) showed that, in two climate models, at least, a significant fraction of the long-term trend (and slow response) was due to increases in convective moistening, which bypasses the TTL cool trap.

We have done analyses testing whether the PDRMIP models and experiments show agreement for the relation between $\Delta SWV_{slow}$ vs the CPT slow response (Figure R1 below). Results from the models and experiments show good agreement. The slope is 0.72 ppmv/K, which is larger than the slope obtained from the fast response. Nevertheless, correlation does not prove causality and this result could arise from either TTL control or if convective moistening also correlates

with the CPT slow response, or some combination. We have added a sentence to the revised manuscript describing this analysis in lines 343-348.

[Figure]

Figure R1: Same as Figure 6a of the submitted manuscript, but for TLS SWV slow response vs the CPT slow response.

*"5. While Fig. 3 shows a consistent relationship between stratospheric water vapor and global mean surface temperature across various forcing, the temperature sensitivity does not seem to be so consistent in Fig. S4. Much more stratospheric moistening is seen in response to the solar forcing than others given the same surface temperature warming. This discrepancy needs to be resolved."*

We list the regression slopes in the unit of ppmv/K in Table S4 and slopes in the unit of %/K in Table S5 in the revised supplement. The slope values in Table S4 are the same as we have shown in Fig. 3 of the submitted manuscript. It may not be clear in Fig. 3 of the submitted manuscript, but it is clear in Table S4 that the sensitivities are indeed larger in some experiments, such as the 2%Solar experiment. This is also the same for slopes in the unit of %/K in Table S5.

*"Line 85-86: How does the averaged of fixed SST with baseline atmosphere compare to the average of the coupled baseline simulations."*

For TLS SWV, the difference between fixed SST baseline simulation and coupled baseline simulation is on the order of $0.01 - 1$ ppmv. For LMS SWV, the difference between fixed SST baseline simulation and coupled baseline simulation is on the order of $0.1 - 1$ ppmv.

The results are averaged over the entire period of the baseline simulations for both fixed SST run and coupled run.

*"Line 96: y=c+ab^x -> y=c+ab^(-x)*

We have updates this (line 119).

*Line 101: Fig. S1 was not showing what is stated here. It seems the intended Fig. S1 is missing.*

*Line 147: Fig. S2-4. -> Fig. S1-3*

*Line 167: Fig. S5 -> Fig. S4"*

We have updated figures and figure numbers in the supplement.

*"Line 191: Does the long wave effect of the tropospheric ozone also contribute?"*

Yes, the tropospheric ozone has the long wave radiative effect. We have edited the text in the revised manuscript in lines 278-280.

**Reviewer #2**

General comments

*1. "The paper largely focuses on interpreting the multi-model responses. While this is of course useful, it stops short of relating the new understanding to any real-world changes in SWV... How much does this work help in understanding past and possible future SWV changes?"*

*"In particular, note there has been some discussion of how the PDRMIP BC perturbations compare to observations (Allan et al, [https://doi.org/10.1038/s41612-019-0073-9)](https://doi.org/10.1038/s41612-019-0073-9)..."*

We have added a new figure (Figure 7) and table (Table 2) and associated discussion to the paper (The "4. Historical changes in SWV" section). In this section (lines 366-400), we use our results to estimate observed changes in SWV and compare those to observations. Our estimate shows reasonable agreement with observed trend over 1980-2010.

*2. "As noted in the specific comments, I feel that there is inadequate recognition that some of the results presented here are also presented, either explicitly or implicitly, in some earlier papers from the PDRMIP group – this is particularly so for the ERFs where no reference to, or comparison with, those earlier results, is given."*

Thanks for pointing this out. We have added references to related results from earlier PDRMIP studies in the revised paper.

Please also see the responses to specific comments below related to previous PDRMIP studies:

*"47-48: There is a slight overlap between this submitted paper and the paper published in ACP by the core PDRMIP team – Hodnebrog et al: https://doi.org/10.5194/acp-19-12887-2019, which is not cited here..."*

Lines 53-55, 94-95, 374-375.

*"129: I think it is necessary that a comparison of ERFs (and the associated feedback parameter) with Richardson et al. (including for the CFCs and N2O in their supplement) is presented both to confirm they are in reasonable agreement and also to make clear that the ERFs derived here are not original work with the PDRMIP output."*

We have added a Table S3 in the revised supplement comparing our ERFs with "ERF$_{sst}$" from Richardson et al. (2019). The comparison shows good agreement. Texts mentioning this comparison are in lines 151-158 of the revised manuscript.

Specific comments

*"14: This conclusion is specific to the TLS"*

Yes, we agree with this, although the cold point temperature does have *some* influence in the lowermost SWV (Dessler et al., 1995). But the control is not as strong as that in the TLS (see Fig. 6b-c in submitted manuscript) and the lowermost SWV is controlled by multiple factors. We have edited the text to make this clearer (lines 13-14).

*"16: "becomes weaker at higher altitudes and at higher latitudes below 150 hPa." This is a bit ambiguous. Does this means heights at pressures below 150 hPa or heights below the height of the 150 hPa surface. These would have opposite meanings."*

It means altitudes below the 150 hPa surface. We have edited the text for clarity (line 16).

*"57: Presumably the 3xCH4 experiments have no resulting change in SWV due to the oxidation of additional methane?"*

Yes, indirect chemical effects are not included in the 3xCH$_4$ experiment. We have added a sentence saying this (line 77-78).

*"90 and many other places: There are repeated statements that there is no surface temperature response in the fixed SST runs, but this is not correct, with implications for the definition of ERF."*

Yes, the reviewer is correct that land surface temperatures can respond to the forcing. We have edited the text in the revised manuscript (lines 108-110).

*"139: tend to be larger" Isn't it clearly larger?*

Yes, the reviewer is correct. We have edited this text in the revised manuscript (line 175).

*"*148 and throughout: Rather little is said about intermodel differences. For example, on HADGEM3, more discussion of its apparent outlier status on some plots seems necessary. The text says it is "likely connected" to the larger surface warming, but it seems the climate sensitivity is about double the multi-model average but the slow SW response is around a factor of 4 larger. Is that because the TTL temperature change is 4 times higher (per unit ERF)?"*

To answer the question about the HadGEM3 model, Figure R2 below shows the equilibrium slow response of TTL temperature ($\Delta T_{slow}$) per unit ERF. The HadGEM3 $\Delta T_{slow}$/ERF is between 2.64-3.97 times the multi-model mean $\Delta T_{slow}$/ERF for experiments 2xCO2, 3xCH4, 2%Solar, 10xBC, and 10xCFC-12. Since the surface warming in HadGEM3 is larger than all other models, its upper tropospheric warming is also largest. Longwave radiation emitted from the upper troposphere warms the TTL level (Lin et al., 2017), so the larger upper tropospheric warming in HadGEM3 also results in larger TTL heating than other models. The relationship between surface warming and TTL warming is not linear.

That said, we cannot conclusively identify a cause given the information archived. So we have removed the claim that the difference is likely connected to surface warming and we have added a sentence saying more work on the causes of these differences is warranted (lines 189-190).

[Figure]

Figure R2: Equilibrium slow response of TTL temperature (100 hPa, averaged between 30°N-30°S) per ERF for all models and perturbations.

*"Another example is that apparently half the models have a slow SW response to BC of the opposite sign (Fig 1a) to the multi-model mean. Is there any obvious reason why? As far as I can see BC causes a warming in all models."*

Figure R4 below shows the vertical profile of tropical temperature slow (a) and fast (b) responses per unit ERF for the 10xBC experiment. The 10xBC does cause a warming at the surface and in the troposphere due to a positive TOA ERF in all models. In the TTL and lower stratosphere (LS), however, the heating is mainly caused by the fast adjustment (Fig. R3b below). The slow temperature response in the TTL is the residual of the total response minus the fast adjustment, which is negative or close to zero (Fig. R3a below).

It is therefore our contention that some of these negative values are artifacts of the method we use to estimate equilibrium response. Support for this comes from Fig. 3 of the paper. The values in this figure come from regressions of $\Delta SWV_{slow}$ vs. $\Delta Ts$ in the BC runs. This method does not require differencing two large numbers, so we feel it is more robust. It shows that most models have a positive response of SWV due to BC-induced warming. For those models that produce negative slopes for $\Delta SWV_{slow}$ vs. $\Delta Ts$ in the BC runs, there is large uncertainty in the regression, because the surface temperature change in those models are small.

We have noted this explanation in the revised manuscript (lines 191-196).

[Figure]

Figure R3: Profiles of equilibrium slow (a) and fast (b) temperature response for the 10xBC experiment, normalized by ERF (K·(Wm$^{-2}$)$^{-1}$), and averaged over 30°N-30°S. The color coding indicates results from different models.

*"One thing I miss from this study, and encourage the authors to look at if they have the resource, is the degree to which the model's background climatology of stratospheric water vapor or TTL temperature could explain some of the intermodel differences."*

We have investigated the SWV in the fixed SST baseline simulations. Based on our analyses, the baseline climatology SWV does not explain the inter-model differences in the responses to forcing agents. As an example, Fig. R4 below shows the TLS SWV slow response (first row) and TTL temperature slow response (second row) vs. the baseline TLS SWV climatology and baseline TTL temperature climatology. We omitted $3xN_2O$, $5xO_3$, and 10xBCSLT, because fewer than three models performed these experiments. There is no correlation between the SWV and temperature slow responses and the baseline climatology. In particular, HadGEM3 produces extremely large slow responses for most experiments, however, in Fig. R4 below, its baseline SWV and temperature climatology is not the largest among the models.

[Figure]

Figure R4: Top row: The TLS SWV slow response (ppmv) vs. the baseline TLS SWV climatology (ppmv). Bottom row: The TTL temperature slow response (K) (100 hPa, averaged over 30°N-30°S) vs. the baseline TTL temperature climatology (K) (100 hPa, averaged over 30°N-30°S). The baseline climatology is obtained from the fixed SST simulations averaged over the last 10 years.

*"156: Is this linear regression done once across all simulations and all perturbations. If not, I am unclear which perturbations have been used for the regression."*

We have added a sentence "We do this regression for each model and perturbation separately" to avoid confusion (lines 202-203).

*"159: This is a relatively short paper and I wondered whether the supplementary figures could be brought into the main text?"*

It remains our opinion that the key figures are included in the paper. Thus, in order to keep the take-home message concise, we have left the content of the supplement unchanged.

*"171-172: This repeats a point already made at 141-142."*

We'll have removed the repeated text.

*"201: I am sorry if I miss it, but I see very little discussion of stratospheric temperature changes in the Jain et al paper. The role of CFCs on the vertical profile of temperature can be seen in many papers such as Forster et al. https://doi.org/10.1007/s003820050182 and Forster and Joshi 10.1007/s10584-005- 5955-7..."*

In the submitted paper where we discussed the radiative heating in the UTLS by CFCs, we were referring to the text in Section 3.3 of (Jain et al., 2000), where they stated that "Halocarbons absorb predominantly in the window region (750-1250 $cm^{-1}$), in the linear line limit; therefore in the stratosphere they absorb the upwelling radiation from the troposphere and increase the heating rate of the stratosphere".

We agree with the reviewer that it is useful to reference papers that explicitly investigated vertical temperature profiles forced by CFCs. We have added these references in the revised manuscript (line 272).

*"\*204: This statement on shortwave radiation is strange. There may be a small shortwave effect from the reduced reflected flux from the troposphere, but there is a long history of simulations that clearly attribute the stratospheric cooling due to increased tropospheric ozone to the decreased upwelling thermal infrared radiation. E.g. Ramaswamy and Bowen https://doi.org/10.1029/94JD01310, Berntsen et al https://doi.org/10.1029/97JD02226 and the Forster et al. paper referred to above."*

Thanks for pointing this out. We have edited the text to say, "Increases of tropospheric $O_3$ ($5xO_3$) reduce the upwelling longwave radiation, which cools the stratosphere. The longwave radiation absorbed heat the TTL region" (lines 278-280). References are also added in lines 278-280.

*"206: "Tropospheric O3 is also transported". As I understand it, ozone is imposed in the models and not advected. I don't know what this sentence means."*

This is correct: In the $5xO_3$ experiment, the PDRMIP group used 5 times the tropospheric ozone distribution (TROP) in the paper by MacIntosh et al. (2016) (line 76). We have removed the text about transport.

*"212: "larger than 50%". CAM5 and MPI-ESM look less than 50%?"*

Yes, this was poorly worded. We have completely re-written the paragraph in lines 274-277.

*"*248: Returning to General Point#1, the Summary feels a very mechanical repetition of the results in the paper without any discussion of the wider implications, remaining uncertainties, or possible future avenues/priorities for improving understanding."*

We added our discussion on wider implications and remaining uncertainties to "4. Historical changes in SWV" section in the revised manuscript (lines 366-400).

*"273: Strictly Fig 5 refers to TLS only"*

We have edited the text (line 427).

*"519-520: I think the markers are only reported when there are more than 3 contributing models?"*

Yes, the multi-model mean and error bars are shown for perturbations that are performed by more than three models. We have added this caveat to the revised figure captions.

*"46L "responses" -> "responds""*

We have modified the text (line 53).

*"Throughout: This may be common usage, but the paper refers throughout to the ensemble mean when other papers would refer to it as the multi-model mean (ensemble could refer to different runs from the same model with perturbed initial conditions or physics…"*

Thanks for pointing this out. To avoid confusion, we have replaced the "ensemble mean" with multi-model mean in the revised manuscript.

**References**

[revised manuscript text omitted]